# MixEval: Deriving Wisdom of the Crowd from LLM Benchmark Mixtures

$^\diamond$**Jinjie Ni**$^{*\dagger}$, $^\diamond$**Fuzhao Xue**$^*$, $^\clubsuit$**Xiang Yue**$^*$,
$^\spadesuit$**Yuntian Deng**, $^\diamond$**Mahir Shah**, $^\diamond$**Kabir Jain**, $^\clubsuit$**Graham Neubig**, $^\diamond$**Yang You**
$^\diamond$National University of Singapore, $^\clubsuit$Carnegie Mellon University, $^\spadesuit$Allen Institute for AI

https://mixeval.github.io/

## Abstract

Evaluating large language models (LLMs) is challenging. Traditional ground-truth-based benchmarks fail to capture the comprehensiveness and nuance of real-world queries, while LLM-as-judge benchmarks suffer from grading biases and limited query quantity. Both of them may also become contaminated over time. User-facing evaluation, such as Chatbot Arena, provides reliable signals but is costly and slow. In this work, we propose `MixEval`, a new paradigm for establishing efficient, gold-standard LLM evaluation by strategically mixing off-the-shelf benchmarks. It bridges (1) comprehensive and well-distributed real-world user queries and (2) efficient and fairly-graded ground-truth-based benchmarks, by matching queries mined from the web with similar queries from existing benchmarks. Based on `MixEval`, we further build `MixEval-Hard`, which offers more room for model improvement. Our benchmarks' advantages lie in (1) a 0.96 model ranking correlation with Chatbot Arena arising from the highly impartial query distribution and grading mechanism, (2) fast, cheap, and reproducible execution (6% of the time and cost of MMLU), and (3) dynamic evaluation enabled by the rapid and stable data update pipeline. We provide extensive meta-evaluation and analysis for our and existing LLM benchmarks to deepen the community's understanding of LLM evaluation and guide future research directions.

## 1 Introduction

**That Which is Measured, Improves.** Evaluation is essential in the AI community for two main reasons: (1) benchmarks provide early signals to model developers, aiding in refining data and model design, and (2) benchmarks guide users in selecting suitable models for specific use cases. Therefore, benchmarks offer feedback to the entire community, facilitating model optimization. Consequently, the main concern of evaluating LLMs is **impartiality**–we need to optimize impartial objectives so that the community advances in the right direction. In practical LLM evaluations, three primary biases contribute to a lack of impartiality: (1) **query bias**–evaluation queries falling short of comprehensiveness or appropriate distribution (2) **grading bias**–the grading process involving significant bias or error (3) **generalization bias**–models overfitting the evaluation data.

**Large Scale User-facing Evaluation Provides a More Impartial Signal.** Practitioners generally adopt either automatic or user-facing approaches for LLM benchmarking. Automatic benchmarking typically employs traditional ground-truth-based benchmarks, such as MMLU [17], which often fail to capture real-world query comprehensiveness and nuance while involving a comparatively impartial grading process; or employs open-ended benchmarks using LLMs as graders, such as MT-Bench [39], suffering from both grading bias and query incomprehensiveness due to the preference biases and high cost of frontier LLM judges. Additionally, the static nature of automatic benchmarks results

---

$^*$Core contributors.
$^\dagger$Correspondence to: Jinjie Ni <jinjieni@nus.edu.sg>

38th Conference on Neural Information Processing Systems (NeurIPS 2024).

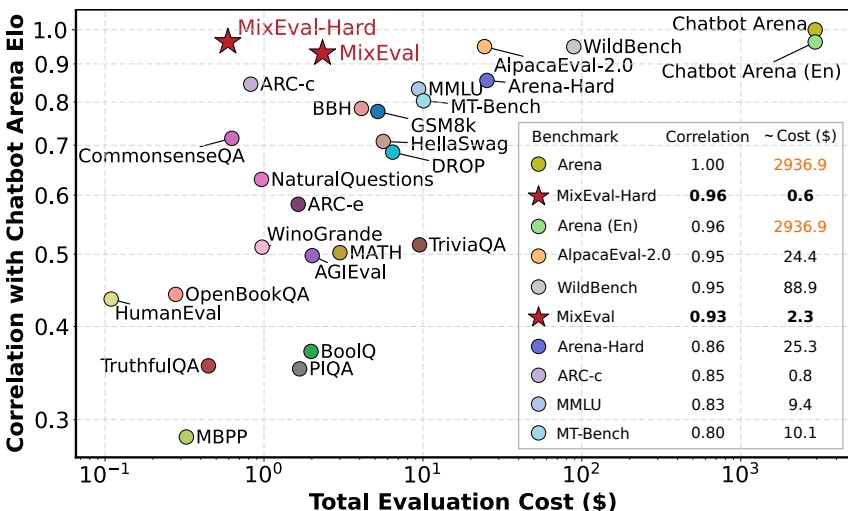

Figure 1: Benchmark correlations (%) with Chatbot Arena Elo, against the total costs of evaluating a single GPT-3.5-Turbo-0125 model. **MixEval** and **MixEval-Hard** show the highest correlations with Arena Elo and Arena Elo (En) among leading benchmarks. We reference the crowdsourcing price for Amazon Mechanical Turk ($0.05 per vote) when estimating the cost of evaluating a single model on Chatbot Arena (approximately $2,936). Chatbot Arena is prohibitively expensive, while **MixEval** and **MixEval-Hard** are cheap and cost-effective alternatives. Details on the correlation and evaluation cost values are provided in Section E.

in contamination over time, amplifying the generalization issue. Such biases lead to significant deviations from gold-standard evaluation, impeding model development. On the other hand, large-scale user-facing benchmarking, such as Chatbot Arena[3] [6], offers more reliable objectives for model development and effectively mitigates the above-mentioned three biases because (1) it collects a vast array of real-world user queries, thereby ensuring superior query comprehensiveness and distribution, (2) it judges diverse and complex model responses stably due to the "wisdom of the crowd" effect [33], where individual judgment noise is averaged out over a large number of samples, mitigating the grading bias, and (3) it continuously receives fresh user queries, mitigating the benchmark contamination issue. Furthermore, it guides model optimization to meet user needs effectively in practical applications, which is a crucial goal of developing models. However, Chatbot Arena is prohibitively expensive (Figure 1), slow, and irreproducible. Moreover, it is not directly accessible for public usage, hindering practitioners from conducting easy and fast model evaluations.

**MixEval: Towards Efficient Gold-Standard LLM Evaluations.** In this work, we aim to establish a highly impartial gold-standard benchmark without compromising efficiency. This can be achieved by leveraging (1) the efficiency and grading impartiality of ground-truth-based benchmarks and (2) the superior comprehensiveness and distribution of real-world user queries. To this end, we propose `MixEval`, a two-stage benchmark reconstruction pipeline consisting of (1) wild query mining and (2) grounding existing benchmarks in the mined queries. We introduce an accurate user query retrieval process, comprising query detection, filtering, and classification. In the detection phase, we train open-source LLMs on self-collected data to detect queries in Common Crawl splits. During filtering, we utilize GPT-4 Turbo to exclude non-query sentences. In classification, we categorize the filtered queries by input and output modalities, retaining text-in-text-out queries for LLM evaluation. To align benchmark queries with real-world queries, we match each crawled web user query with its most similar query in the benchmark pool and the corresponding ground truth answer. We designate the resulting benchmark as `MixEval`. To improve the benchmark's ability to distinguish strong models, we derive a challenging subset from `MixEval`, termed `MixEval-Hard`. To mitigate the overfitting issue, we periodically update the data points in `MixEval` and `MixEval-Hard` using our fast, stable pipeline, which performs benchmark mixture with a different batch of wild queries from

---

[3]The Chatbot Arena leaderboard is not the sole indicator of real-world human preferences, but it currently serves as one of the gold standards within the community. Therefore, we utilize it as a reliable source of approximation.

the same distribution, showing low model score variance (0.36 Std. on a 0-100 scale) and significant version difference (85% unique query ratio). We thereby effectively mitigate the above-mentioned three evaluation biases through the proposed benchmark mixture pipeline, while maintaining high efficiency. As shown in Figure 1, `MixEval` and `MixEval-Hard` achieve similar model rankings as Chatbot Arena while being far less costly.

**Why use MixEval?** `MixEval` offers five significant advantages for practitioners: (1) **accurate** model ranking, demonstrated by a 0.96 correlation with Chatbot Arena, (2) **fast**, **cheap** and **reproducible** execution, requiring only 6% the time and cost of MMLU and with no dependence on human input, (3) **dynamic** benchmarking enabled by low-effort and stable updating mechanism, (4) a **comprehensive** and **less biased** query distribution, as it bases queries on a large-scale web corpus, and (5) a **fair** grading process without preference bias, ensured by its ground-truth-based nature.

**Research Contributions**

- We developed a pipeline for detecting real-world instructions, capable of mining queries to build benchmarks and providing a scalable solution for collecting vast amounts of real-world instruction-following data.

- We introduced a new way to utilize benchmarks, demonstrating that real-world query distributions and user preferences can be reconstructed by strategically mixing off-the-shelf benchmarks with web-mined queries.

- To the best of our knowledge, `MixEval` creates the first ground-truth-based dynamic benchmark with general-domain queries, benefiting from a rapid and stable data updating mechanism.

- The resulting dynamic benchmarks, *i.e.,* `MixEval` and `MixEval-Hard`, exhibit significant correlations (0.93 and 0.96) with real-world user preference leaderboard (*i.e.,* Chatbot Arena) and showcase high impartiality and efficiency.

- We provide meta-evaluation and extensive analysis for `MixEval` and other leading LLM benchmarks, delivering detailed insights that enhance the community's understanding of LLM evaluation and guide future research directions.

## 2 LLM Benchmarks are Biased from Realistic User Queries and Preferences

**How Much Do Our Benchmarks Reflect Real-world User Queries and Preferences?** The rapid advancement of LLMs has led to the introduction of numerous benchmarks. However, the community may still lack a comprehensive understanding of how well these benchmarks align with real-world use cases and human preferences. Without such understanding, the signals derived from evaluations might be misleading, thereby impeding model development. To investigate this issue, we (1) analyze the correlations between benchmarks (Figures 1 and 10) and (2) visualize their query distributions in a unified 2-D space (Figure 2). The setups for these analyses are detailed in Section E.

### 2.1 Important Takeaways

**Current benchmarks show a limited correlation with human preferences.** Figures 1 and 10 reveal that many benchmarks exhibit low correlation with human preferences (Arena Elo). The highest correlations are Arena-Hard (0.86) for LLM-judged and ARC-c (0.85) for ground-truth-based benchmarks. Using the benchmark mixture technique, our `MixEval`

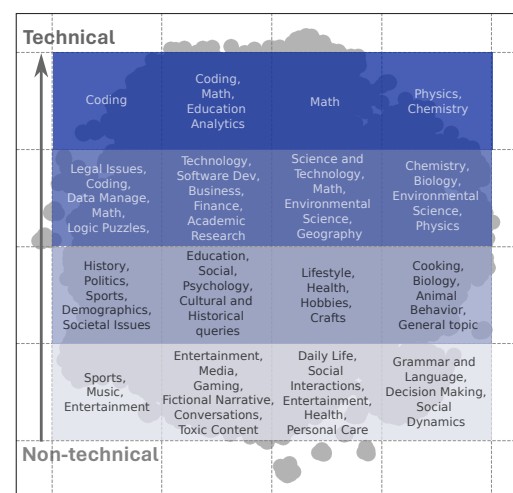

Figure 3: Query topic summarization for Figure 2. The plot aggregates all queries and divides them into 16 regions. From each region, 100 queries are uniformly sampled and analyzed by GPT-4 for topic summarization. A clear trend is observed, with topics transitioning from non-technical at the bottom to technical at the top.

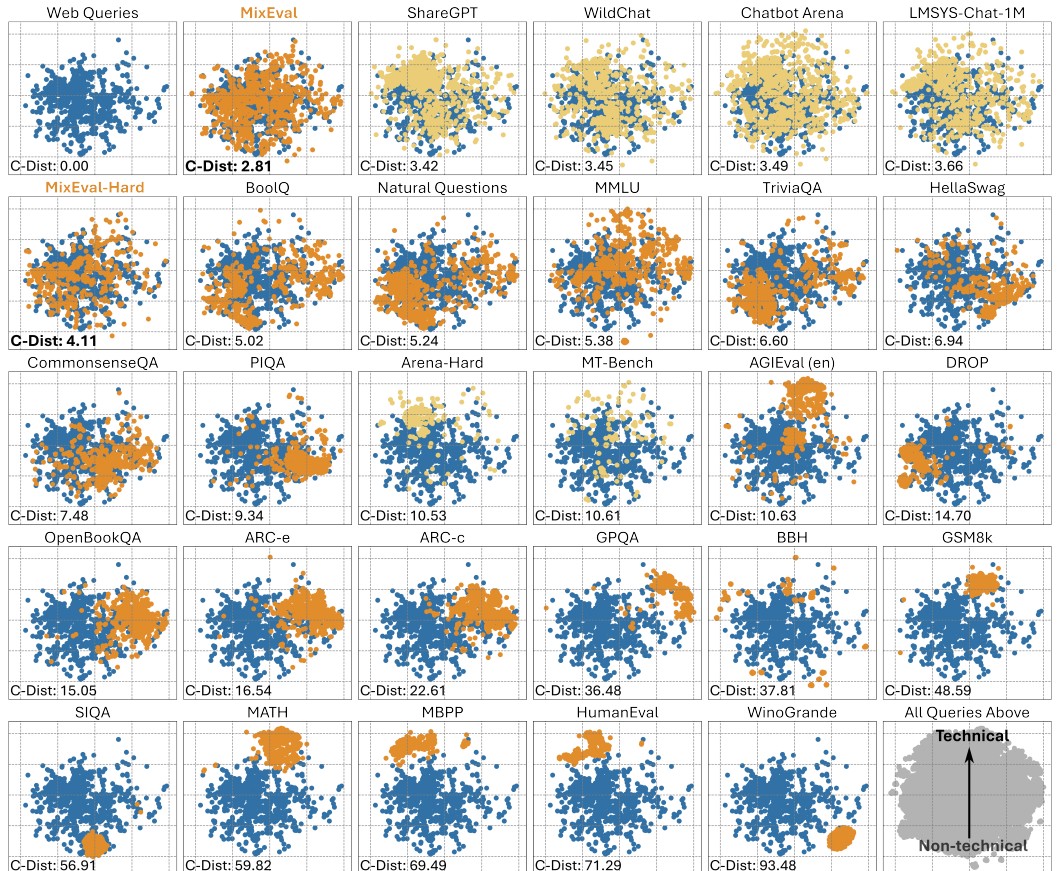

Figure 2: Query Topic Distribution of the Benchmarks. Ground-truth-based benchmarks are represented by orange dots, wild datasets by yellow dots, and LLM-judged benchmarks (MT-Bench and Arena-Hard) by yellow dots, all plotted against our detected web queries shown as blue dots. Query sentence embeddings were dimensionally reduced to map them onto a unified 2-D space, facilitating direct comparisons of topic distributions across benchmarks. As we move from the bottom to the top of the figure, query topics transition from non-technical to technical. Topic summaries for each region are detailed in Figure 3.

and `MixEval-Hard` achieve the highest correlations with human preferences, at 0.93 and 0.96, respectively.

**Most benchmarks exhibit a skewed topic distribution.** Ground-truth-based and LLM-judged benchmarks show a skewed query distribution compared to detected web queries and wild datasets. Notably, BoolQ, Natural Questions, and MMLU align more closely with wild user queries due to their data collection methods. Specifically, questions in BoolQ and Natural Questions originate from Google Search, while MMLU is designed to cover a wide range of topics (57 topics, including Atari games [2]).

**Query comprehensiveness is crucial.** General-domain benchmarks, which are not tailored for specific domains in their data collection pipelines, exhibit a stronger correlation with Arena Elo than domain-specific ones. Notably, 10 out of 13 general-domain benchmarks have an Arena Elo correlation score above 0.5, whereas only 1 out of 8 domain-specific benchmarks achieves this. This underscores the significance of comprehensive query topics for achieving high correlation with human preferences.

**Some general-domain benchmarks are actually domain-specific.** Despite being labeled as general-domain benchmarks, DROP and WinoGrande have limited scopes, often narrower than many domain-specific benchmarks. As depicted in Figure 3, DROP queries mainly address History, Politics, Sports,

Demographics, and Societal Issues, whereas WinoGrande queries focus primarily on Grammar, Language, Decision Making, and Social Dynamics.

**User population size affects the query distribution.** Figure 2 shows distribution differences in wild queries across varying user population sizes. ShareGPT, grounded in 100 million[4] active users of ChatGPT by mid-2023, contrasts with WildChat [38], Chatbot Arena Conversations [6], and LMSYS-Chat-1M [39], which have user bases of 0.2 million, 0.13 million, and 0.21 million, respectively. The global internet user count was 5.4 billion[5] in 2023, an order of magnitude larger than all considered wild datasets. Consequently, user bases of the internet, ShareGPT, and other datasets span three distinct orders of magnitude. ShareGPT's larger user population (second order of magnitude) yields a distribution most similar to web queries from the global internet user base (third order of magnitude), both visually and in cluster distance (C-Dist).

**Chatbot Arena and Arena-Hard queries exhibit biases.** Compared to web queries and ShareGPT data, datasets from the Chatbot Arena website—Chatbot Arena Conversations and LMSYS-Chat-1M—have a higher proportion of technical queries (as presented in Figure 3, queries with higher position on the map are more technical). This indicates a user base skewed towards technical users, potentially affecting evaluation results, as an effective LLM benchmark should mimic real-world use cases. Furthermore, a minor discrepancy exists between web and ShareGPT queries, suggesting that one or both of them may still slightly deviate from actual real-world query distributions. Moreover, Arena-Hard queries exhibit a pronounced bias towards technical topics. This likely stems from the design of their data pipeline for sampling hard prompts. We will demonstrate that employing a carefully designed sampling technique is essential to preserve the query distribution while enhancing difficulty. This is supported by `MixEval-Hard`'s similar distribution to the original web queries and wild datasets (see Section 3.3).

# 3 MixEval

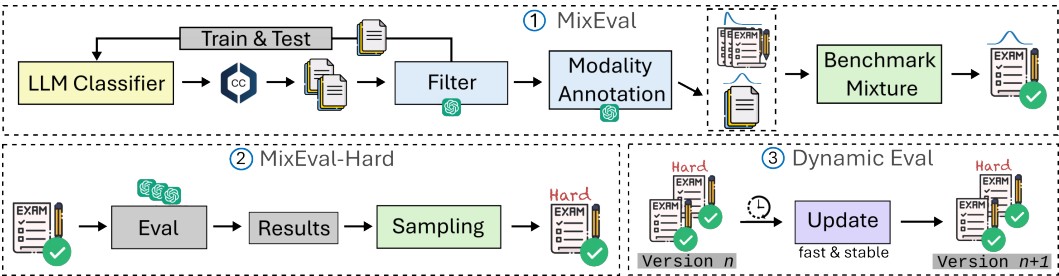

Figure 4: `MixEval`, a two-stage benchmark reconstruction pipeline, comprises (1) web query detection and (2) benchmark mixture. We further introduce `MixEval-Hard` to enhance model separability, alongside a dynamic updating mechanism to mitigate contamination risk.

In Section 2, we show that current ground-truth-based and LLM-judged benchmarks have skewed query distributions and limited correlation with human preferences. Additionally, LLM-judged benchmarks suffer from LLM preference bias and both of them become contaminated over time. In contrast, Chatbot Arena is less biased and more dynamic but requires slow and expensive human preference data collection, resulting in irreproducible outcomes.

To address these issues, we introduce `MixEval` (Figure 4), which aligns ground-truth-based LLM benchmarks with real-world user queries. This method uses user queries mined from the web and matches them with similar queries from existing benchmarks, and involves two stages: (1) user query detection from the web and (2) benchmark mixture. To improve model separability and reduce contamination, we also propose `MixEval-Hard` and a dynamic updating mechanism.

## 3.1 Web User Query Detection

In this stage, we detect user queries from Common Crawl [12]. Both recall and precision are crucial to ensure the query distribution reflects real-world scenarios. Therefore, we developed two benchmarks

---

[4]https://www.mlyearning.org/chatgpt-statistics
[5]https://www.statista.com/statistics/273018/number-of-internet-users-worldwide/

to evaluate our query detector's performance. The first benchmark includes self-collected in-the-wild user queries as positive samples, with non-query datasets such as Wikipedia [16] as negative samples. The second, higher-quality benchmark contains positive and negative samples hand-picked by our authors from in-the-wild query and non-query datasets. In preliminary experiments, direct prompting of open-source language models performed poorly on our benchmarks. Thus, we developed a rectification pipeline to ensure high recall and precision cost-effectively. We started with a detection phase to gather training data. Testing various open-source LLMs, Vicuna 33B [7] achieved a high recall (>99%) on our test sets with careful prompt engineering, ensuring that very few positive samples were missed initially. In this phase, we detected around 20k queries using Vicuna 33B over a subset of Common Crawl. We then used GPT-4 to more accurately label these data as positive or negative samples, and used the resulting data to train Vicuna 33B. The trained Vicuna 33B achieved high recall (>99%) and precision (>98%) on our benchmarks and detected 2M user queries from the entire Common Crawl. Finally, we prompted GPT-4 Turbo to classify them, extracting text-in-text-out queries for LLM evaluation. Future work will address queries with other I/O modalities.

## 3.2 Benchmark Mixture

To bridge wild user queries $\mathcal{Q}$ and ground-truth LLM benchmarks, we create a benchmark pool $\mathcal{B} = \{\mathcal{B}_1, \mathcal{B}_2, ..., \mathcal{B}_n\}$, where each $\mathcal{B}_n = \{b_1, b_2, ..., b_k\}$ represents a distinct ground-truth LLM benchmark. We define a mapping $f : q_i \mapsto b_j$, with $q_i \in \mathcal{Q}$ and $b_j \in \mathcal{B}$. For each $q_i \in \mathcal{Q}$, we rank similarities between each $(q_i, b_j)$ pair and select the most similar $b_j$ that satisfies $\theta$: $b_j = f(q_i) = \arg\max_{b_j \in \mathcal{B}} S(q_i, b_j)$ s.t. $\theta$. We use the dot-product between normalized sentence embeddings as the similarity score $S(\cdot)$. When retrieving the top-1 $b_j$, $\theta$ is a length constraint on the input (or context) field of each $b_j$, addressing the effect of long inputs in the benchmark data mixture. The sentence embeddings of queries are computed using the `all-mpnet-base-v2` model from SentenceTransformers [23]. To ensure quality and comprehensive sample coverage, we selected the development and test splits of widely adopted benchmarks from diverse domains and topics. Details and distributions of these benchmarks are provided in Appendix D.

## 3.3 MixEval-Hard

Table 1: The key statistics of `MixEval` and `MixEval-Hard`. With dynamic benchmarking, the numbers may vary slightly while the number of queries will not change.

| | # Queries | Avg. # Toks per Query | Avg. # Inputs | Avg. # Toks per Input | Min # Toks per Input | Max # Toks per Input | English Ratio | Eval Type |
|---|---|---|---|---|---|---|---|---|
| **MixEval** | 4000 | 23 | 0.3 | 41.3 | 6 | 954 | 95.15% | Ground Truth |
| **MixEval-Hard** | 1000 | 27.3 | 0.4 | 47.3 | 7 | 954 | 95.22% | |

Frontier LLMs are rapidly approaching human-level performance across diverse tasks. As these models progress, existing benchmarks will become saturated, hindering differentiation between models. Although `MixEval` reflects typical user queries, it is constrained by the benchmark pool $\mathcal{B}$'s overall difficulty. Our results in Table 3 indicate that top models, such as GPT-4 Turbo and Claude 3 Opus, have surpassed 88% accuracy on `MixEval`. To improve the benchmark's ability to discriminate between very strong models, we extract a challenging subset from `MixEval` to create `MixEval-Hard`.

Given `MixEval` denoted as $\mathcal{B}'$, we sample a hard subset $\mathcal{B}''$ from $\mathcal{B}'$ by computing a difficulty score $\xi_i$ for each entry, prioritizing higher scores. Consider a set of model prediction results $\mathcal{A}$, where $\mathcal{A}$ is a 0-1 matrix of shape $(N_{model}, N_{\mathcal{B}'})$, with 1 indicating an incorrect model response. Here, $N_{model}$ is the number of models, and $N_{\mathcal{B}'}$ is the number of questions in $\mathcal{B}'$. The difficulty score $\xi_i$ for a query $b_i'$ is computed by $\xi_i = \vec{\mu} \cdot \vec{\mathcal{A}}_i$, where each model's result on question $i$ is weighted by its accuracy $\mu_j$ on the dataset. Given $\xi = \{\xi_1, \xi_2, ..., \xi_{N_{\mathcal{B}'}}\}$, we sample from $\mathcal{B}'$ with rejection:

$$\mathcal{B}'' = \{b_i' \in \mathcal{B}' : p(b_i') \text{ and } \alpha(\mathcal{B}'' \cup \{b_i'\}, \mathcal{B}') \leq \tau\},$$

where $\alpha(x, y)$ denotes the cluster distance between $x$ and $y$. The probability of drawing $b_i'$, $p(b_i') = \frac{e^{\lambda \xi_i}}{\sum_{b_k' \in \mathcal{B}'} e^{\lambda \xi_k}}$, is based on $\xi_i$. This rejection sampling ensures that `MixEval-Hard` is difficulty-first

while maintaining a balanced query distribution. We obtain 1000 samples for `MixEval-Hard`. The statistics of `MixEval` and `MixEval-Hard` are detailed in Table 1.

## 3.4  Dynamic Benchmarking

Table 2: Stability test for dynamic benchmarking. Five models tested across five updated versions of `MixEval` show an average mean of 77.64 and a Std. of 0.36, validating the stability of model scores over versions. The unique web query ratio, averaged across all version pairs, is 99.71%, and the unique benchmark query ratio is 85.05%, indicating significant differences between versions.

| | GPT-3.5-Turbo-0125 | GPT-3.5-Turbo-1106 | Claude 3 Haiku | Mistral-Small | Reka Edge | Avg. | Unique Web Query Ratio | Unique MixEval Query Ratio |
|---|---|---|---|---|---|---|---|---|
| **Mean** | 79.66 | 79.25 | 80.32 | 80.57 | 68.42 | 77.64 | 99.71% | 85.05% |
| **Std.** | 0.26 | 0.28 | 0.34 | 0.56 | 0.35 | 0.36 | | |

Static benchmarks risk contamination over time as models may overfit to the benchmark data [6, 32, 36], undermining evaluation reliability. To address this, we periodically update the data points in `MixEval` and `MixEval-Hard` using the automatic pipeline described above, i.e., performing benchmark mixtures based on the queries uniformly sampled from the massive web queries detected, which completes updates within one minute. Table 2 shows score stability and version differences. We created five versions of `MixEval` by altering the random seed when sampling web queries and ran five models on them. As shown, the average mean and standard deviation (Std.) for the models across the versions are 77.64 and 0.36, respectively, demonstrating high score stability. For each pair of versions, we compute the unique sample ratio for sampled web queries and benchmark data points. Given samples $X = \{x_1, x_2, ..., x_n\}$ from version A and $Y = \{y_1, y_2, ..., y_n\}$ from version B, the unique sample ratio $\mathcal{R}$ is calculated as $\mathcal{R} = \frac{|X-Y|+|Y-X|}{X \cup Y}$, representing the unique ratio of the $X \cup Y$ set. The average unique web query ratio across all version pairs is 99.71%, and the unique ratio for `MixEval` versions is 85.05%, indicating significant differences between versions. This efficient updating mechanism, alongside stable model scores and significant data point variations, effectively mitigates benchmark contamination. Additionally, we plan to dynamically expand our benchmark pool with newly released benchmarks to further enhance the mixed benchmark distribution.

To summarize, we update the data points of MixEval via (1) batch web query update (sampling different web queries batches from the crawled web queries), (2) source web query update (updating all the web queries with the latest Common Crawl) or (3) benchmark pool update (incorporating new ground-truth-based benchmarks to the benchmark pool). Since the mechanism of MixEval is to match web queries with benchmark pool samples, the above three updating methods refreshes both the web queries (the first and the second method) and benchmark pool samples (the third method).

## 4  Results

### 4.1  Experiment Settings

We evaluate models on `MixEval` and `MixEval-Hard` using the Transformers library [31] for open-source models, adhering to the official settings in their Hugging Face model card. Proprietary models are assessed via their official API endpoints, using the latest versions as of April 30, 2024. Chat models employ official chat templates or FastChat chat templates [39], and base models are evaluated in a 5-shot setting. Both `MixEval` and `MixEval-Hard`, comprising samples from various benchmarks, demonstrate the inadequacies of traditional rule-based parsing methods across all benchmarks and models. To improve parsing accuracy, we use GPT-3.5-Turbo-0125 as the model parser to either score the response (free-form questions) or extract the model's choice (multiple-choice problems). The stability of the GPT-3.5 Turbo parser is evidenced in Table 2 of this paper and Table 4 of [37]. We will also provide an open-source model parser with its stability test to ensure long-term reproducibility. Section J details the model parser prompts, and Section I compares the model parser to the rule parser. Models are evaluated on 4 or 8 A100 GPUs. All correlations with Arena Elo are based on the Chatbot Arena Leaderboard as of May 1, 2024.

## 4.2 Effectiveness of MixEval

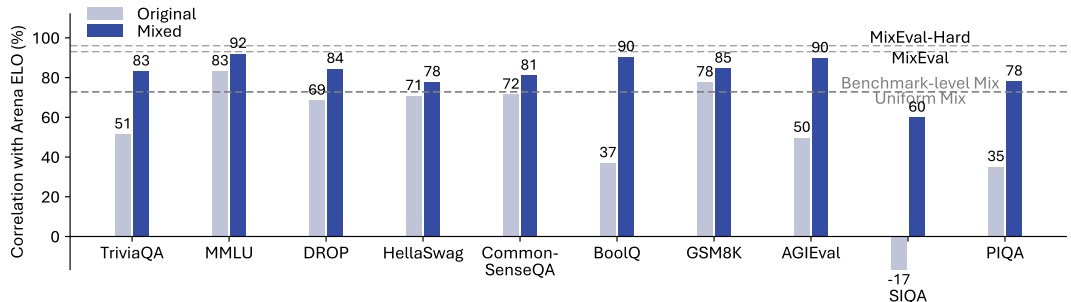

Figure 5: Our approach improves the correlation with Arena Elo and Arena Elo (En) (Figure 12) for all the main splits of `MixEval` and outperforms benchmark-level and uniform mixture.

**MixEval and MixEval-Hard achieve the highest correlation with Arena Elo and Arena Elo (En) among all benchmarks.** As shown in Figures 1 and 10, `MixEval` and `MixEval-Hard`, derived from the proposed `MixEval` pipeline to simulate diverse user queries, achieve significantly higher correlations (10% higher than the top SOTA benchmark) with human preferences (both Arena Elo and Arena Elo (En)), ranking second and first, respectively. Notably, `MixEval-Hard`'s correlation with Arena Elo is even slightly higher than with Arena Elo (En). As discussed in Section 4.5, query difficulty impacts human preference correlation. Therefore, `MixEval-Hard`'s superior correlation may partially result from increased query difficulty. The high correlations of `MixEval` and `MixEval-Hard` with human preferences enable efficient, cost-effective, and reliable model ranking compared to human-in-the-loop benchmarks.

**MixEval improves the correlation with Arena Elo and Arena Elo (En) across all main benchmark splits of MixEval.** In Figure 5, we select the top-10 benchmarks from our pool with sufficient sample sizes (see sample number distribution in Figure 8). For each benchmark, we present (1) the correlation between Arena Elo and the original benchmark, and (2) the correlation between Arena Elo and the `MixEval`-mixed version. Remarkably, **all** benchmarks exhibit significant improvements in their correlations with Arena Elo after being processed by `MixEval`. The correlation increase is notably high (>40%) in benchmarks such as BoolQ, AGIEval, SIQA, and PIQA. `MixEval` and `MixEval-Hard`, which aggregate all benchmarks, consistently outperform any individual benchmark mixture, underscoring the importance of a large benchmark pool and query comprehensiveness.

**MixEval outperforms both benchmark-level and uniform mixtures.** Figure 5 illustrates the correlations with Arena Elo for benchmark-level and uniform mixtures. The benchmark-level mixture samples questions uniformly from each benchmark, proportional to its split size in `MixEval`. The uniform mixture samples an equal number of questions from all benchmarks. Both methods yield significantly lower human preference correlations than `MixEval` and `MixEval-Hard`. Furthermore, the benchmark-level mixture offers negligible improvement over the uniform mixture. These findings underscore the importance of an appropriate sample-level mixture, as implemented by `MixEval`.

**MixEval effectively maps real-world user queries to ground-truth-based benchmarks.** Figure 2 shows the query distributions of leading benchmarks. Both `MixEval` and `MixEval-Hard` closely resemble web queries and popular wild datasets, highlighting `MixEval`'s efficacy in aligning benchmark query distributions with real-world data. The maps in Figure 2 are ordered by their cluster distances to our identified web queries, showing that wild datasets align more closely with our web queries than other LLM benchmarks. This underscores the robustness of our web query detection pipeline and the solid grounding of `MixEval`. Additionally, as discussed in Section 2, ShareGPT, with a larger user base (100M) compared to other wild datasets (0.1M-0.2M), shows the highest similarity to our web queries, which are based on a global internet user population (5.4B), further validating the accuracy of our web query detection.

### 4.3 Evaluation Results

**Leaderboard** Table 3 (Section G) presents the detailed evaluation results on `MixEval`, `MixEval-Hard`, and their main subsets. Claude 3 Opus and GPT-4 Turbo consistently achieve

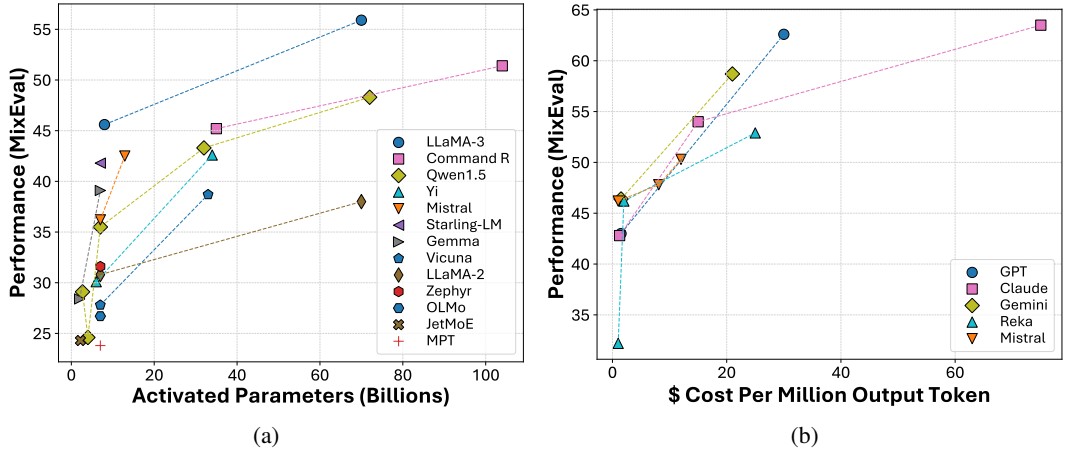

(a)                                                        (b)

Figure 6: Activated parameters and API price per performance of open-source and proprietary models.

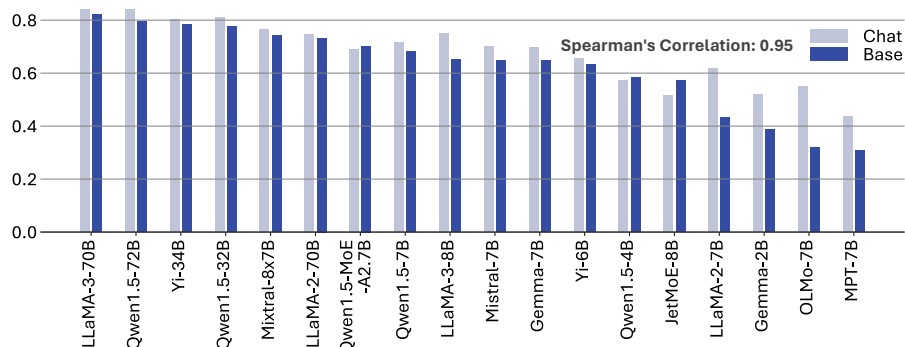

Figure 7: The performance of chat and base models of the same model series in Table 3. Chat and base model scores show a high correlation.

the highest performance across nearly all splits, except for BoolQ-Mixed. Gemini 1.5 Pro ranks third on both `MixEval` and `MixEval-Hard`, followed closely by Claude 3 Sonnet, LLaMA-3-70B-Instruct, and Reka Core, with similar scores. Notably, all these frontier models, except LLaMA-3-70B-Instruct, support multi-modal input understanding. The LLaMA-3-8B-Instruct model is the top-performing 7B model, outperforming some of the latest large models, such as Command R (35B) and Qwen1.5-32B-Chat (32B). Proprietary models generally outperform open-source models.

**Cost-effectiveness** Figure 6 compares the models in Table 3 in terms of cost-effectiveness. Figure 6a examines the relationship between activated parameters and performance for open-source LLMs, while Figure 6b compares API price against performance for frontier proprietary LLMs. Both figures exhibit a roughly log-linear relationship between performance and the x-axis metric. In Figure 6a, the Llama-3 series stands out as the most performant and parameter-efficient among open-source models. The MoE models, such as Mixtral-8x7B-Instruct-v0.1, Qwen1.5-MoE-A2.7B-Chat, and JetMoE-8B-Chat, demonstrate superior parameter efficiency. The proprietary data points reveal a clearer log-linear pattern. GPT-4 Turbo is more cost-effective than Claude 3 Opus, offering comparable performance at less than half the price. The Gemini series exhibits similar cost-effectiveness to the GPT series, while the Reka series parallels the cost-effectiveness of the Claude series. We conduct detailed error analysis in Section H to compare error rates of open-source and proprietary models on different `MixEval` splits. We also showcase error responses of frontier models in Section H.1 to identify their potential weaknesses.

### 4.4 Can We Approximate the Human Preferences for Base Models?

The crowdsourced evaluation of LLMs, based on human preferences, assesses two main aspects: (1) model capability, optimized mainly during pre-training, and (2) non-capability attributes like toxicity and helpfulness, refined during post-training. We explore whether human preferences for models can be predicted before post-training, leading to the question: which stage, pre-training or post-training, more significantly influences human preferences in crowdsourced LLM evaluations? We evaluated the base versions of the model series in Table 3. Notably, the correlations in Figure 7 show a 0.95 correlation between base and chat models, indicating `MixEval`'s potential to approximate human preferences pre-post-training. This implies that pre-training may have a greater impact on crowdsourced LLM evaluations, with post-training minimally altering human preference rankings. However, we also observe that the post-training has more impact on some smaller models, all of which went through heavy supervised post-training.

### 4.5 What Affects the Correlations with Human Preference?

**Comprehensiveness and other features, such as difficulty, impact correlation.** As shown in Figure 10, general-domain benchmarks typically exhibit a higher correlation with human preference compared to domain-specific benchmarks, highlighting the importance of query comprehensiveness. However, comprehensiveness is not the sole factor. Three observations support this: (1) Benchmarks like GSM8K, despite their skewed distributions (Figure 2), achieve a high correlation (0.78) with human preference, while others with high topic overlap with real-world queries, such as BoolQ, achieve a low correlation (0.37). (2) ARC-e and ARC-c, despite similar topic distributions, show significantly different correlations (Figure 10), likely due to varying difficulty levels. This indicates that other query features, such as difficulty, are critical to correlation with human preference. (3) As shown in Figure 5, `MixEval` increases the correlation for each individual benchmark through benchmark mixture. For an individual benchmark, the topic becomes less comprehensive post-mixture since the mixed version represents a subset of the original; thus, the correlation gain is not due to a more comprehensive topic distribution. These observations suggest that correlation gains with human preference are influenced by factors beyond solely comprehensiveness, possibly including query difficulty and topic density. Section F presents the full correlation matrix and analyzes other factors affecting benchmark correlations.

## 5 Conclusion

In this paper, we present `MixEval`, an approach that bridges real-world queries and ground-truth-based evaluation by mining user queries from the web and matching them with similar benchmark queries. `MixEval` and its hard variant can offer accurate evaluations that highly align with Chatbot Arena. `MixEval` operates locally and rapidly, eliminating the need for slow, costly human preference data collection or biased model judgment. `MixEval`'s data points can be stably updated within one minute, mitigating benchmark contamination. We thereby effectively mitigate the query, grading, and generalization biases in LLM evaluation through the proposed benchmark mixture pipeline, while maintaining high efficiency. Our meta-evaluation and extensive analysis of `MixEval` and other popular LLM benchmarks demonstrate `MixEval`'s effectiveness, providing insights to enhance the community's understanding of LLM evaluation.

## Acknowledgement

We thank Yao Fu, Balázs Galambosi, Jason Phang, Jason Wei, Piotr Nawrot, Luca Soldaini, Guanzhi Wang, Deepanway Ghosal, Bo Li, Junhao Zhang, Yifan Song, Zangwei Zheng, Zian Zheng, Qinghong Lin, Wenhu Chen, Bill Yuchen Lin, and colleagues from CMU NeuLab for insightful discussions and pointers.

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

# A  Frequently Asked Questions

We list the potentially frequently asked questions and the point-to-point answers as follows:

## A.1  Why are real-world human queries and preferences important?

One of the primary applications for AI model development is the automation of complex tasks traditionally performed by humans. As AI models coexist with humans, frequent interactions with humans are necessary to manage these tasks effectively. These interactions predominantly involve natural language queries, as it is the most common medium for human communication. Given the significance of human interaction in the use cases for general AI models, it is crucial to evaluate AI models—particularly large language models (LLMs) that rely on natural language—under conditions that mirror real-world scenarios, *i.e.,* receiving human queries and assessing performance based on human preferences. Evaluating models based on their real-world use cases is well-supported across various research disciplines [3, 8, 20].

## A.2  Why do you use web queries as real-world user queries?

Because web queries are grounded in the largest human population (5.4 billion internet users) among the accessible query sources. Figure 2 illustrates the query distribution differences in wild queries across various user population sizes. ShareGPT, with 100 million[6] active users by mid-2023, contrasts with WildChat [38], Chatbot Arena Conversations [6], and LMSYS-Chat-1M [39], which have user bases of 0.2 million, 0.13 million, and 0.21 million, respectively. The global internet user count was 5.4 billion[7] in 2023, an order of magnitude larger than all considered wild datasets. Consequently, the user bases of the internet, ShareGPT, and other datasets span three distinct orders of magnitude. ShareGPT's larger user population (second order of magnitude) yields a distribution most similar to web queries from the global internet user base (third order of magnitude), both visually and in cluster distance (C-Dist), validating that user population size affects query distribution. Compared to web queries and ShareGPT data, datasets from the Chatbot Arena website—Chatbot Arena Conversations and LMSYS-Chat-1M—have a higher proportion of technical queries (as presented in Figure 3, queries with higher position on the map are more technical). This indicates a user base skewed towards technical users, potentially affecting evaluation results, as an effective LLM benchmark should mimic real-world use cases.

## A.3  Why do you use benchmark mixtures instead of training judge models directly on Arena Conversations to achieve a similar model ranking with Arena Elo?

The crucial difference lies in the scoring methods: using ground truth answers versus LLM judges. Ground-truth-based evaluation is more interpretable, faster, and cost-effective compared to LLM-as-judges. Furthermore, training effective judge models is highly challenging because (1) LLMs possess inherent preference biases [39], and (2) to evaluate other models accurately without ground truth answers, the judge model must be either superior to or at least on par with the models it assesses. Consequently, a large model is required, complicating the training process.

## A.4  Why isn't the cluster distribution ranking introduced in Figure 2 consistent with the rankings in Figures 1 or 10?

The key to human preference correlation, as indicated in Figure 5, lies in ensuring that a benchmark's query distribution aligns with a subset of the wild queries, rather than encompassing the entire wild query distribution. This is evidenced by the high correlation between the `MixEval`-mixed domain-specific benchmarks and Arena Elo in Figure 5. However, aligning with only a subset of wild queries significantly impacts the cluster distance metric shown in Figure 2. Notably, covering all regions of wild queries enhances correlation scores, as demonstrated by `MixEval` achieving higher correlation in Figure 5.

---

[6]https://www.mlyearning.org/chatgpt-statistics
[7]https://www.statista.com/statistics/273018/number-of-internet-users-worldwide/

### A.5 How long does it take to dynamically update MixEval-Hard?

The update of `MixEval-Hard` is somewhat slower than `MixEval`, which can be updated within 1 minute. `MixEval-Hard`, a subset of `MixEval`, is sampled based on the prediction results of several models. Thus, the update time depends on the models used to sample this subset. If only GPT-4 Turbo's prediction results are used to rank the question difficulties, the total update time is approximately 2 minutes, which remains rapid. However, according to [22], GPT-4 Turbo may yield a lower score in this condition compared to using results from multiple models.

### A.6 Arena-Hard's correlation with Arena Elo (En) is reported to be 0.94 in their blog while 0.83 in Figures 1 and 10

The number for Arena-Hard in Figures 1 and 10 is derived from the model scores reported on Arena-Hard's leaderboard as of May 01, 2024, using the Arena Elo (En) values from the same date. This discrepancy may arise because the figure reported in Arena-Hard's blog did not account for all models listed on their leaderboard. The number of models used for each pair of benchmarks shown in Figure 10 is reported in Figure 11

### A.7 Why GPQA is not included in Figures 1 and 10?

Because we didn't find enough data points for GPQA that share enough common models with other benchmarks.

### A.8 Is MixEval totally unbiased?

No, `MixEval` is not entirely unbiased. This is due to several factors: the detection pipeline is not perfectly accurate, Common Crawl data collection introduces biases, and there are inherent biases from web users in the real world. However, `MixEval` is relatively less biased because it draws from a broad internet user base. This is supported by: (1) The maps in Figure 2, which are ordered by their cluster distances to our identified web queries, indicate that wild datasets align more closely with our web queries than with other LLM benchmarks. This demonstrates the robustness of our web query detection pipeline and the solid grounding of `MixEval`. (2) As discussed in Section 2, ShareGPT, with its extensive user base (100M) compared to other wild datasets (0.1M-0.2M), shows the highest similarity to our web queries, which are based on the global internet user population (5.4B). This further validates the accuracy of our web query detection. (3) The trained web query detector achieved high recall (>99%) and precision (>98%) on our internal web query detection benchmarks.

### A.9 Will the pipeline that creates MixEval-Hard introduce some noise that influences the result?

`MixEval-Hard` sampling relies on the difficulty scores of benchmark questions, which inherently include some dataset annotation errors. During our error case study (Section H), we identified several annotation issues. However, the number of annotation errors was minimal, rendering the noise negligible. Furthermore, the high correlation with Arena Elo demonstrates that the introduced noise does not significantly affect the model rankings.

## B  Related Work

**LLM Benchmarking**

Both frontier and open-source LLMs have made significant strides in recent years. Evaluation scores are a core objective in LLM development, necessitating an effective evaluation pipeline for successful model advancement. Current LLM evaluation methods can be categorized into three main types: (1) ground-truth-based evaluation, (2) LLM-as-judge evaluation, and (3) crowdsourcing human-preference-based evaluation.

Ground-truth-based evaluation, or closed-ended evaluation, involves ranking the outputs of base and chat LLMs against predefined correct answers. Various benchmarks have been introduced by

the research community for this purpose [1, 4, 5, 9–11, 14, 17, 21, 24–26, 28, 29, 35, 40]. These benchmarks facilitate rapid and straightforward LLM evaluation, providing clear and unbiased answer judgments due to their closed-ended nature. However, ground-truth-based benchmarks often exhibit topic bias (as illustrated in Figure 2) and may not accurately represent the diversity of real-world user queries, limiting their ability to assess the nuanced capabilities of LLMs.

On the other hand, two other categories of evaluation approaches primarily focus on the open-ended evaluations of chat LLMs. The LLM-as-judge evaluation uses frontier models to rank the responses to a set of open-ended queries without ground-truths. These queries are either manually designed [39] or sourced from crowdsourcing platforms [6]. However, due to the high cost of using frontier models as judges, such approaches are not scalable to a large number of user queries. This limitation hinders the ability to reflect the complexity and diversity of real-world queries and may deviate from the true distribution (see Figure 2). Additionally, previous research has identified several biases in frontier model judges, including verbosity bias, position bias, and self-enhancement bias [39]. These biases can lead to unfair model rankings in practical evaluations. Additionally, the static nature of both ground-truth-based and LLM-as-judge benchmarks results in contamination over time, diminishing the reliability of evaluation outcomes.

As a comparison, Chatbot Arena [6] serves as a robust benchmark for evaluating chat LLMs. It operates as a benchmarking platform where anonymous, randomized battles are conducted in a crowdsourced environment. The platform's extensive real-world user queries provide comprehensive and less biased evaluations, ensuring the accuracy and stability of model rankings. Additionally, its real-time nature prevents models from overfitting the benchmark, thereby avoiding contamination issues. However, obtaining a stable model score requires more than 5,000 human interactions and several days, making the process labor-intensive and slow. Furthermore, its open-ended format limits its ability to evaluate base models. Many open-ended queries, such as creating a travel plan, lack definitive standards for distinguishing good from bad answers [30], resulting in evaluations that are not purely ability-oriented.

**Web Query Detection**

Currently, real-world text-in-text-out user queries are primarily sourced from chat platforms [6, 27, 38, 39]. Our concurrent work, MAmmoTH2 [34], also identifies real-world user queries from the web. However, MAmmoTH2 has fundamentally different objectives compared to `MixEval`. MAmmoTH2 focuses on detecting large-scale domain-specific query-answer pairs, while `MixEval` targets general-purpose user queries that accurately reflect the real-world user query distribution. This difference in objectives results in distinct web query detection pipelines.

## C   Considerations of Web User Query Crawling

Our user queries are not directly crawled from the web; instead, they are identified using Common Crawl, an openly available corpus of web crawl data widely used in the research community. Furthermore, we do not release the raw detected queries, while only releasing the final mixed version of `MixEval` for two reasons: (1) the raw detected queries may contain toxic content or unexpected sensitive information, and (2) we update our benchmarks dynamically to avoid contamination. Releasing the detected raw queries would make the dynamic benchmarking process more predictable, reducing its effectiveness.

## D   Additional Benchmark Statistics

To ensure quality and comprehensive sample coverage, we selected the development and test splits of widely adopted benchmarks encompassing various domains and topics.

- General-domain benchmarks: MMLU [17], BoolQ [9], HellaSwag [35], ARC [10], CommonSenseQA [29], AGIEval [40], OpenbookQA [21], GPQA [24], WinoGrande [25], TriviaQA [19], DROP [14], and BBH [28].
- Domain-specific benchmarks: Math: GSM8K [24] and MATH [18]; Coding: MBPP [1] and HumanEval [5]; Physics: PIQA [4]; and Social Interactions: SIQA [26].

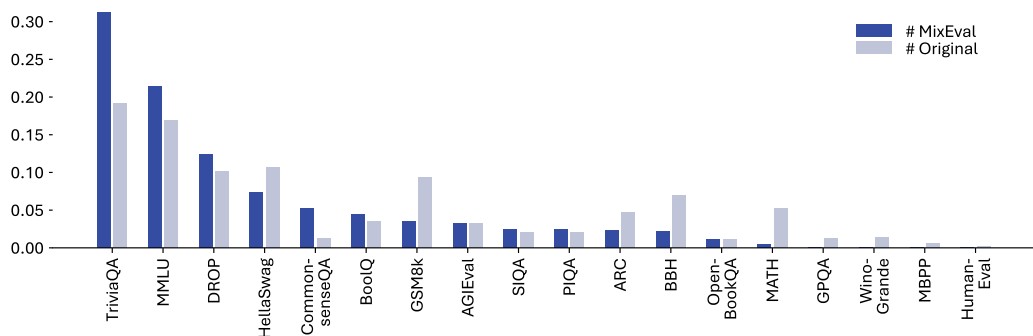

Figure 8: The normalized number of queries in `MixEval` and the original benchmarks.

According to Figure 2, the mixed benchmark `MixEval` exhibits the highest overlap with $\mathcal{Q}$ among all benchmarks, suggesting that $\mathcal{B}$ adequately represents the wild query distribution. Let $\mathcal{B}' = \{\mathcal{B}'_1, \mathcal{B}'_2, ..., \mathcal{B}'_n\}$ denote the mixed benchmark. The distributions of $\mathcal{B}$ and $\mathcal{B}'$ are illustrated in Figure 8. A positive correlation between the sizes of the mixed and original benchmarks is observed. Intuitively, a larger benchmark is likely to be retrieved more frequently; however, this is not universally true. Benchmarks with skewed sample distributions, such as HellaSwag, GSM8k, ARC, BBH, MATH, and GPQA, have a smaller relative size after mixing. This indicates that both quantity and distribution influence how frequently a benchmark is retrieved by $\mathcal{Q}$. Overall, the retrieved benchmark splits exhibit a long-tail distribution.

## E  Implementation details for Benchmark Correlation Matrix, Query Distribution, and Evaluation Cost

**Correlation Matrix Heatmap (Figures 1 and 10).** We present the correlation matrix of prominent benchmarks, where warmer colors indicate higher correlations. Model scores are collected from various sources, including the Chatbot Arena Leaderboard [6], Open LLM Leaderboard [15], and OpenCompass Leaderboard [13]. Our data collection adheres to three principles: (1) We exclude scores reported by model authors, relying solely on evaluation leaderboards to ensure fairness. (2) For each benchmark, scores are sourced from a single platform to eliminate the influence of varying evaluation settings on model rankings. (3) When multiple sources are available for a benchmark, we select the one with the highest number of models in common with other benchmarks. The number of common models for each pair of benchmarks is detailed in Figure 11.

**Query Distribution Map (Figure 2).** We present the distribution of benchmark queries sorted by their distance to our detected web queries. Each benchmark (orange or yellow) is plotted against the detected wild queries (blue). We uniformly sampled 1000 queries from each LLM benchmark and wild dataset, with a sampling number of 200 for MT-Bench and Arena-Hard due to their smaller sizes. We combined the query embeddings and reduced their dimensions to the same 2-D space to facilitate direct comparisons of the benchmark query distributions. A detailed case study revealed that the reduced space primarily represents the topics of the queries, with queries on similar topics clustering in specific regions of the map. To better understand the topic distribution of different benchmarks, we divided the map into 16 patches based on location (Figure 3). We then uniformly sampled 100 queries from each patch and used GPT-4 to summarize the topics of the sampled queries. As illustrated in Figure 3, the 2-D query distribution exhibits a distinct regional trend: queries located higher on the map are more technical. The distribution transitions from non-technical topics, such as Social Interactions, at the bottom to technical ones, such as Programming and Mathematics, at the top.

**Evaluation Cost Estimation.** As illustrated in Figure 9, we consider two costs when evaluating the performance of GPT-3.5-Turbo-0125 on each benchmark: the inference cost and the judging (scoring) cost. The inference cost computation for ground-truth-based and LLM-as-judge benchmarks are straightforward, involving only the estimation of model input and output tokens for each benchmark. We estimate the model output tokens to be 20 for ground-truth-based benchmarks and 329 for open-

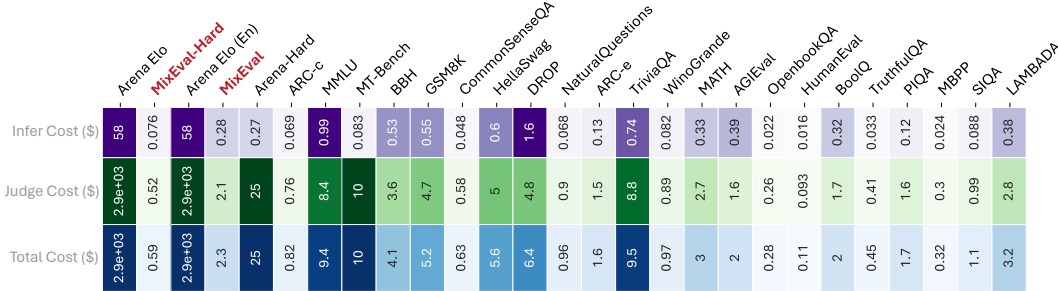

| | Arena Elo | MixEval-Hard | Arena Elo (En) | MixEval | Arena-Hard | ARC-c | MMLU | MT-Bench | BBH | GSM8K | CommonSenseQA | HellaSwag | DROP | NaturalQuestions | ARC-e | TriviaQA | WinoGrande | MATH | AGIEval | OpenbookQA | HumanEval | BoolQ | TruthfulQA | PIQA | MBPP | SIQA | LAMBADA |
|---|---|---|---|---|---|---|---|---|---|---|---|---|---|---|---|---|---|---|---|---|---|---|---|---|---|---|---|
| Infer Cost ($) | 58 | 0.076 | 58 | 0.28 | 0.27 | 0.069 | 0.99 | 0.083 | 0.53 | 0.55 | 0.048 | 0.6 | 1.6 | 0.068 | 0.13 | 0.74 | 0.082 | 0.33 | 0.39 | 0.022 | 0.016 | 0.32 | 0.033 | 0.12 | 0.024 | 0.088 | 0.38 |
| Judge Cost ($) | 2.9e+03 | 0.52 | 2.9e+03 | 2.1 | 25 | 0.76 | 8.4 | 10 | 3.6 | 4.7 | 0.58 | 5 | 4.8 | 0.9 | 1.5 | 8.8 | 0.89 | 2.7 | 1.6 | 0.26 | 0.093 | 1.7 | 0.41 | 1.6 | 0.3 | 0.99 | 2.8 |
| Total Cost ($) | 2.9e+03 | 0.59 | 2.9e+03 | 2.3 | 25 | 0.82 | 9.4 | 10 | 4.1 | 5.2 | 0.63 | 5.6 | 6.4 | 0.96 | 1.6 | 9.5 | 0.97 | 3 | 2 | 0.28 | 0.11 | 2 | 0.45 | 1.7 | 0.32 | 1.1 | 3.2 |

Figure 9: Evaluation cost breakdown for the cost estimation in Figure 1. The total evaluation cost is broken down into the inference cost and judge cost.

ended benchmarks [8]. To compute the evaluation cost of GPT-3.5-Turbo-0125 on the Chatbot Arena, we use the voting number of GPT-3.5-Turbo-0125 on the Chatbot Arena leaderboard as its query count, with each query's token count estimated from the Chatbot Arena Conversations dataset. Since models on Chatbot Arena are evaluated pairwise, both input and output tokens are doubled. The judging cost[9] for ground-truth-based benchmarks is estimated similarly, accounting for the input and output tokens of the model parser. However, estimating the human judgment cost for Chatbot Arena is more complex. We reference the crowdsourcing price for Amazon Mechanical Turk (MTurk), specifically the rate for a Facebook Account Holder[10] ($0.05 per vote). Under this pricing scheme, evaluating a single model on Chatbot Arena costs approximately $2936, making it a highly expensive process.

## F  What Affects the Correlations with Human Preference and Other Benchmarks?

**Both comprehensiveness and other features, such as difficulty, make an impact.** As shown in Figure 10, general-domain benchmarks tend to achieve a higher correlation with human preference compared to domain-specific benchmarks, underscoring the significance of benchmark query comprehensiveness. However, comprehensiveness is not the only factor that matters. Three observations support this: (1) Some benchmarks with highly skewed distributions, such as GSM8K, achieve a promising correlation (0.78) with human preference, while others with high topic overlap with real-world queries, like BoolQ, achieve a low correlation (0.37). (2) Despite their similar topic distributions, ARC-e and ARC-c show significantly different correlations with human preference (Figure 10), likely due to their differing difficulty levels, as they are known to have different levels of difficulty. This illustrates that other important query features, such as difficulty, are not apparent on the 2-D map that primarily features topic distribution but are crucial to the correlation with human preference. (3) As shown in Figure 5, `MixEval` increases the correlation with human preference for each individual benchmark through benchmark mixture. For an individual benchmark, the topic becomes less comprehensive after the mixture because the mixed version represents a subset of the original on the topic map; thus, in this case, the correlation gain does not stem from a more comprehensive topic distribution. These three observations suggest that the human preference correlation gain is not solely related to topic comprehensiveness; other features, such as query difficulty and topic density, also play a crucial role in improving correlation scores.

**Benchmarks that are highly correlated with human preferences also tend to be correlated with each other, whereas those that are less correlated with human preferences are similarly less correlated with most other benchmarks.** The heat map reveals a consistent red region in the top-left, signifying high correlation, while the rest of the map is predominantly blue and inconsistent, indicating low correlation. This suggests that model rankings on benchmarks closely aligned with

---

[8]According to the averaged output token for GPT-3.5-Turbo-0125 as presented at: https://lmsys.org/blog/2024-04-19-arena-hard/.

[9]The judge cost for Arena-Hard and MT-Bench is directly taken from https://lmsys.org/blog/2024-04-19-arena-hard/.

[10]https://requester.mturk.com/pricing

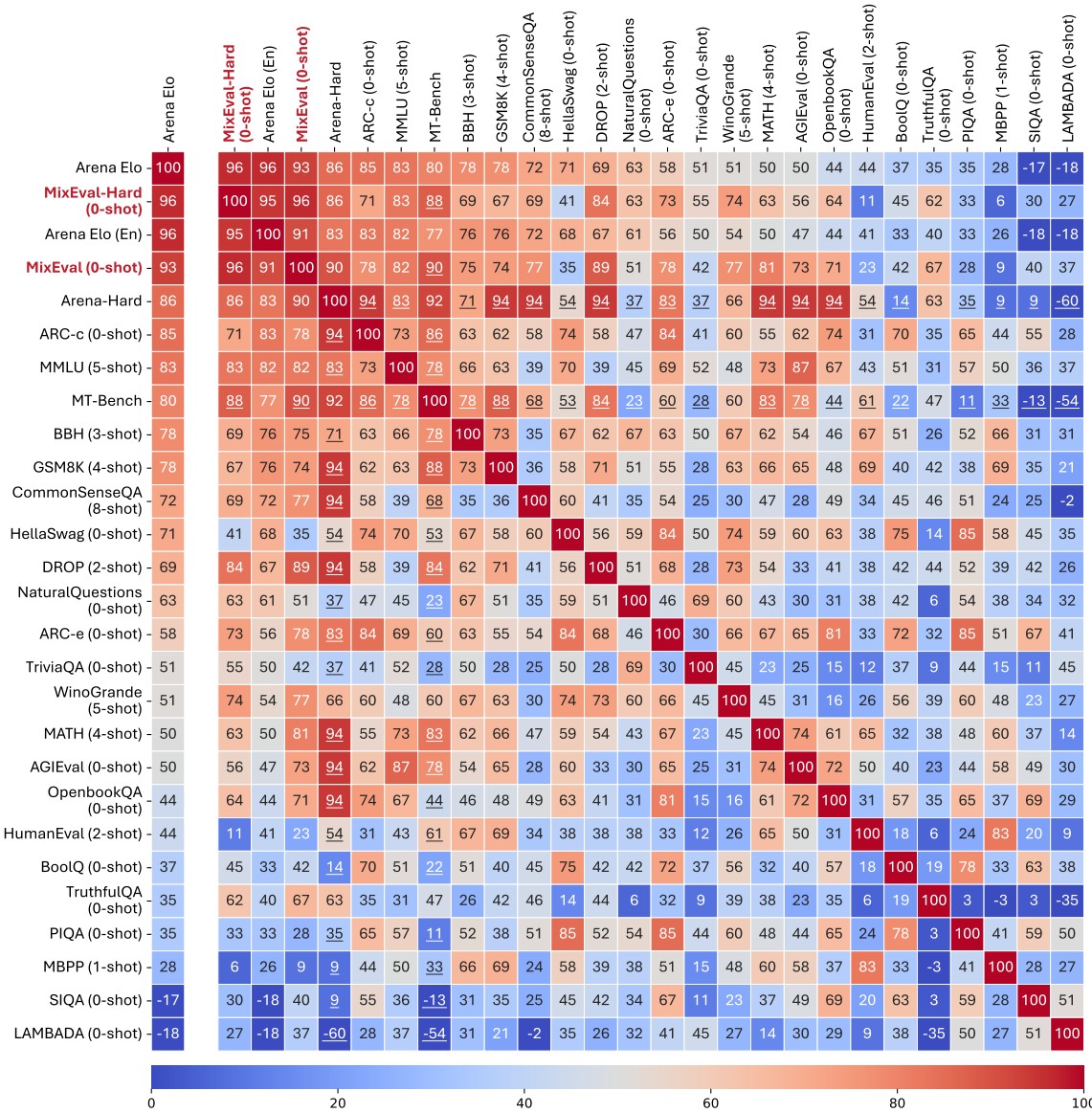

Figure 10: The correlation matrix for benchmarks. `MixEval` and `MixEval-Hard` achieve the highest correlations with Chatbot Arena Elo. Each value of the heatmap represents the Spearman's rank correlation (%) between the model rankings of the corresponding benchmark pairs, where a warmer color indicates a higher correlation and a cooler color indicates a lower correlation. The underlined numbers indicate the data for the corresponding benchmark pairs are insufficient (<15 models). The detailed statistics on the number of models used for each pair of benchmarks are presented in Figure 11.

human preferences are more stable and reflect a "True" ranking, whereas the remaining benchmarks exhibit greater variability in model rankings.

**Benchmarks within the same domain exhibit higher correlations.** Despite a low correlation with human preferences, some domain-specific benchmarks demonstrate a relatively high correlation with other benchmarks in the same domain. For instance, MBPP shows a correlation of only 0.28 with human preference but a substantial 0.83 with HumanEval. Similarly, MATH has a correlation of 0.50 with human preference yet presents a 0.66 correlation with GSM8K. Furthermore, ARC-e has a correlation of 0.58 with human preference while achieving a notable 0.84 correlation with ARC-c.

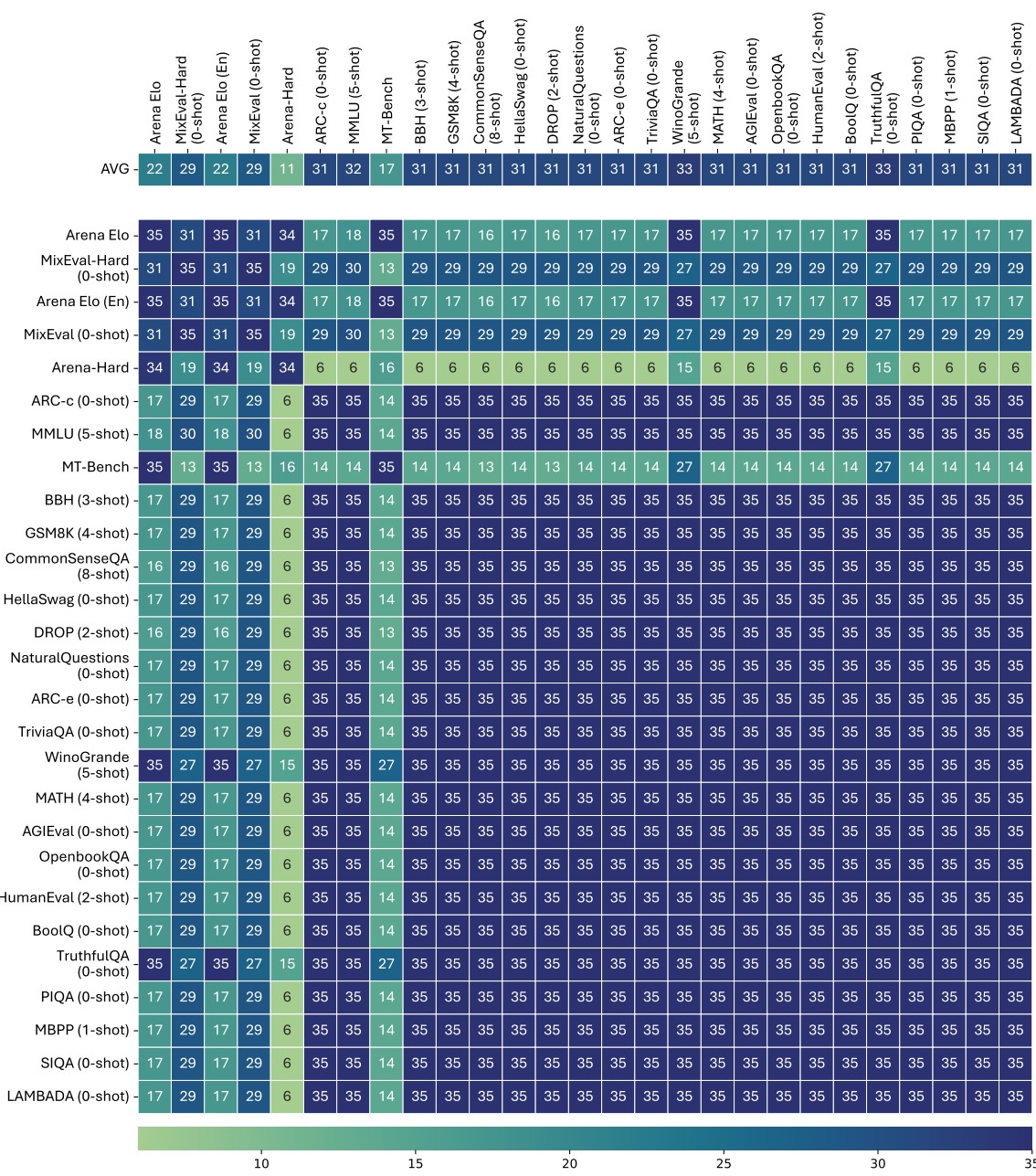

Figure 11: The number of models used for each pair of benchmarks shown in Figure 10.

## G   Detailed Evaluation Results

Table 3 presents our evaluation results on `MixEval`, `MixEval-Hard`, and their main subsets. Overall, Claude 3 Opus and GPT-4 Turbo consistently achieve the highest performance across nearly all splits, except for the BoolQ-Mixed split. Gemini 1.5 Pro ranks third on both `MixEval-Hard` and `MixEval`, followed closely by Claude 3 Sonnet, LLaMA-3-70B-Instruct, and Reka Core, with comparable scores. Notably, all these frontier models, except LLaMA-3-70B-Instruct, support multi-modal input understanding. The LLaMA-3-8B-Instruct model is the top-performing 7B model, outperforming some of the latest large models, such as Command R (35B) and Qwen1.5-32B-Chat (32B). In general, proprietary models outperform open-source models.

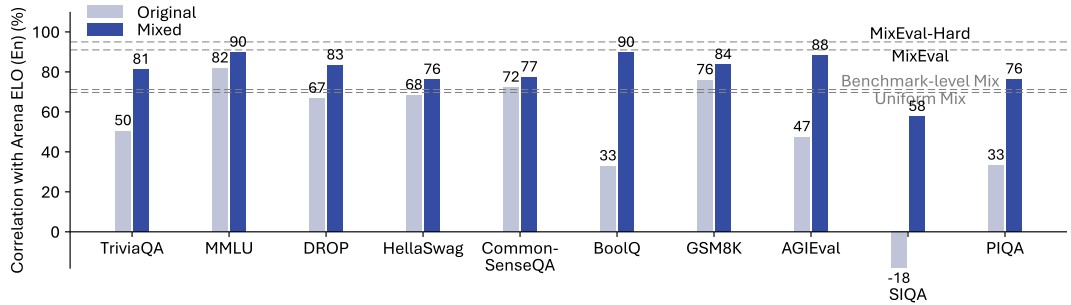

Figure 12: Our approach improves the correlation with Arena Elo and Arena Elo (En) for all the main splits of `MixEval` and outperforms benchmark-level and uniform mixture.

Table 3: The Evaluation results of chat models on `MixEval`, `MixEval-Hard`, and their sub-splits.

| | Mix Eval -Hard | Mix- Eval | TriviaQA- Mixed | MMLU- Mixed | DROP- Mixed | HellaSwag- Mixed | Common senseQA- Mixed | BoolQ- Mixed | TriviaQA- Hard- Mixed | MMLU- Hard- Mixed | DROP- Hard- Mixed |
|---|---|---|---|---|---|---|---|---|---|---|---|
| Proportion | 100% | 100% | 31.2% | 21.4% | 12.4% | 7.4% | 5.3% | 4.4% | 26.6% | 23.1% | 16.7% |
| Claude 3 Opus | **63.5** | 88.1 | 90.4 | **83.2** | **91.5** | **93.3** | **87.7** | 85.1 | 71.4 | **55.0** | **75.2** |
| GPT-4 Turbo | 62.6 | **88.8** | **91.2** | 82.8 | 91.0 | 92.6 | 85.4 | 88.0 | **73.1** | 45.5 | 71.0 |
| Gemini 1.5 Pro | 58.7 | 84.2 | 85.3 | 79.2 | 84.2 | 89.2 | 84.4 | 87.4 | 67.8 | 44.6 | 64.8 |
| LLaMA-3-70B-Instruct | 55.9 | 84.0 | 83.1 | 80.5 | 90.1 | 81.8 | 83.0 | **88.6** | 60.5 | 46.3 | 74.5 |
| Claude 3 Sonnet | 54.0 | 81.7 | 84.2 | 74.7 | 87.7 | 85.9 | 82.5 | 85.1 | 59.1 | 40.7 | 66.9 |
| Reka Core | 52.9 | 83.3 | 82.8 | 79.3 | 88.1 | 88.6 | 81.6 | 85.7 | 51.6 | 46.3 | 66.6 |
| Command R+ | 51.4 | 81.5 | 83.3 | 78.9 | 80.4 | 83.5 | 82.1 | 86.9 | 57.5 | 42.0 | 65.0 |
| Mistral-Large | 50.3 | 84.2 | 88.3 | 80.2 | 88.6 | 65.0 | 83.5 | 87.4 | 55.5 | 42.4 | 61.6 |
| Qwen1.5-72B-Chat | 48.3 | 84.1 | 83.9 | 80.1 | 85.1 | 87.9 | 86.3 | 85.7 | 49.9 | 37.7 | 56.5 |
| Mistral-Medium | 47.8 | 81.9 | 86.8 | 76.3 | 83.2 | 72.4 | 82.5 | 86.3 | 59.8 | 38.5 | 47.1 |
| Gemini 1.0 Pro | 46.4 | 78.9 | 81.0 | 74.9 | 82.6 | 74.7 | 80.2 | 86.3 | 58.2 | 35.5 | 54.1 |
| Mistral-Small | 46.2 | 81.2 | 85.1 | 75.2 | 86.1 | 73.4 | 77.8 | 81.7 | 56.0 | 33.8 | 52.6 |
| Reka Flash | 46.2 | 79.8 | 76.4 | 75.4 | 86.7 | 90.6 | 80.7 | 85.7 | 42.9 | 34.6 | 65.0 |
| LLaMA-3-8B-Instruct | 45.6 | 75.0 | 71.7 | 71.9 | 86.4 | 65.7 | 78.3 | 86.9 | 40.2 | 40.7 | 67.6 |
| Command R | 45.2 | 77.0 | 80.9 | 75.0 | 72.0 | 75.8 | 77.4 | 84.0 | 57.0 | 39.0 | 42.0 |
| Qwen1.5-32B-Chat | 43.3 | 81.0 | 75.7 | 78.0 | 82.9 | 85.9 | 88.2 | 85.1 | 39.1 | 29.9 | 54.4 |
| GPT-3.5-Turbo | 43.0 | 79.7 | 85.2 | 74.5 | 84.8 | 63.0 | 81.6 | 85.1 | 46.4 | 35.1 | 55.4 |
| Claude 3 Haiku | 42.8 | 79.7 | 79.9 | 76.1 | 85.0 | 75.8 | 78.8 | 86.9 | 42.4 | 30.7 | 51.5 |
| Yi-34B-Chat | 42.6 | 80.1 | 82.7 | 73.6 | 86.1 | 86.9 | 78.8 | 80.6 | 41.5 | 29.9 | 57.1 |
| Mixtral-8x7B-Instruct-v0.1 | 42.5 | 76.4 | 82.5 | 72.0 | 79.5 | 54.2 | 77.4 | 82.9 | 48.5 | 37.2 | 47.7 |
| Starling-LM-7B-beta | 41.8 | 74.8 | 75.1 | 69.0 | 86.4 | 48.5 | 84.9 | 84.6 | 33.4 | 34.2 | 62.9 |
| Gemma-1.1-7B-IT | 39.1 | 69.6 | 64.3 | 66.9 | 80.6 | 66.3 | 73.6 | 80.6 | 30.3 | 39.0 | 55.1 |
| Vicuna-33B-v1.3 | 38.7 | 66.3 | 79.2 | 59.2 | 71.4 | 30.3 | 61.8 | 73.7 | 42.5 | 39.4 | 36.6 |
| LLaMA-2-70B-Chat | 38.0 | 74.6 | 80.0 | 69.8 | 79.8 | 67.3 | 74.1 | 76.6 | 42.2 | 27.7 | 42.2 |
| Mistral-7B-Instruct-v0.2 | 36.2 | 70.0 | 73.7 | 67.3 | 72.8 | 54.2 | 66.0 | 77.7 | 33.5 | 29.4 | 44.3 |
| Qwen1.5-7B-Chat | 35.5 | 71.4 | 64.1 | 68.7 | 76.4 | 76.1 | 82.1 | 81.7 | 29.0 | 29.0 | 50.0 |
| Reka Edge | 32.2 | 68.5 | 60.0 | 63.6 | 80.0 | 74.7 | 80.7 | 77.1 | 18.6 | 26.4 | 56.9 |
| Zephyr-7B-$\beta$ | 31.6 | 69.1 | 74.7 | 64.9 | 77.3 | 39.1 | 69.3 | 78.3 | 30.2 | 24.2 | 45.3 |
| LLaMA-2-7B-Chat | 30.8 | 61.7 | 68.8 | 59.4 | 69.3 | 35.7 | 61.3 | 62.9 | 24.8 | 30.3 | 44.3 |
| Yi-6B-Chat | 30.1 | 65.6 | 66.1 | 65.4 | 70.5 | 52.5 | 69.8 | 69.7 | 18.9 | 26.8 | 43.7 |
| Qwen1.5-MoE-2.7B-Chat | 29.1 | 69.1 | 65.9 | 69.5 | 64.6 | 72.7 | 81.1 | 74.9 | 21.9 | 26.8 | 39.5 |
| Gemma-1.1-2B-IT | 28.4 | 51.9 | 53.7 | 51.5 | 59.8 | 26.6 | 57.1 | 60.0 | 31.9 | 30.3 | 27.8 |
| Vicuna-7B-v1.5 | 27.8 | 60.3 | 66.4 | 58.7 | 68.3 | 24.9 | 62.7 | 65.7 | 25.9 | 23.4 | 33.2 |
| OLMo-7B-Instruct | 26.7 | 55.0 | 51.7 | 57.1 | 53.1 | 55.9 | 64.6 | 71.4 | 24.7 | 27.3 | 22.9 |
| Qwen1.5-4B-Chat | 24.6 | 57.2 | 46.0 | 61.4 | 57.2 | 54.9 | 74.1 | 68.0 | 16.5 | 17.3 | 28.6 |
| JetMoE-8B-Chat | 24.3 | 51.6 | 46.8 | 58.5 | 27.0 | 86.2 | 68.4 | 64.6 | 19.2 | 25.5 | 11.5 |
| MPT-7B-Chat | 23.8 | 43.8 | 50.2 | 37.8 | 50.0 | 25.6 | 36.3 | 60.0 | 17.5 | 24.7 | 31.0 |

# H  Error Analysis

Figure 13 illustrates the error rates of the models evaluated on the main splits of `MixEval`. We separately compute the error rates for proprietary and open-source models to facilitate comparison. Both model types exhibit significant errors on the AGIEval split of `MixEval`, underscoring its difficulty. In contrast, performance on the PIQA split is generally saturated. Notably, there is a substantial performance gap between proprietary and open-source models on the GSM8K split, with considerable gaps also observed on the HellaSwag, TriviaQA, and DROP splits.

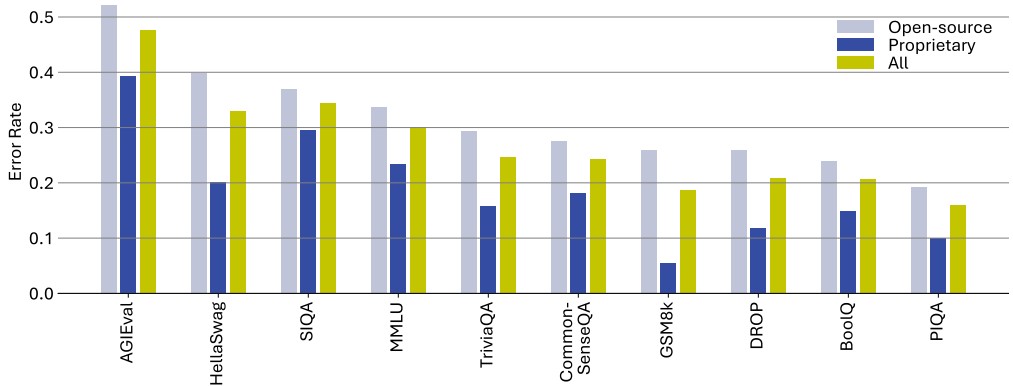

Figure 13: Averaged error rates of open-source, proprietary, and all models on `MixEval` splits.

In Section H.1, we conduct case studies to examine the error cases made by frontier proprietary models. For each case, we present the incorrect responses from each model. We identify three primary causes of confusion for these models: strong domain knowledge, complex reasoning, and vague question definitions. Additionally, we identify several annotation issues within current benchmarks, though these are negligible in number.

## H.1 Error Cases

---

**Error Case 1**

**Question:**
Letter to the editor: You say that if the government were to confiscate a portion of the wages of convicted burglars when they reenter the workforce, it would be a form of stealing, hence an abuse of power. Yet under the proposal now being considered, the government would confiscate such wages in order to fund an account to compensate burglary victims. So even if confiscating a portion of burglars' wages were a form of stealing, it would still be justified.
Which one of the following principles, if valid, most helps to support the argument in the letter to the editor?
A. Money stolen from a burglar should be given to that burglar's victims.
B. Burglars are obligated to provide compensation to the same individuals they victimized.
C. The motive prompting an action determines whether or not that action is justified.
D. A crime is justified only if it is a means of compensating people who deserve compensation.
E. Stealing is never justified even if it benefits someone who has been a burglary victim.
Answer with the option letter from the given choices directly.
**Answer:** C
**Split:** AGIEval
**Main Causes of Errors:** Hard Reasoning

**Wrong Answers:**

`Claude 3 Haiku`: The principle that most helps to support the argument in the letter to the editor is (B) Burglars are obligated to provide compensation to the same individuals they victimized.

The letter argues that even if confiscating a portion of burglars' wages is a form of stealing, it would still be justified because the purpose is to fund an account to compensate burglary victims. This suggests that the burglars have an obligation to provide compensation to the individuals they victimized, which aligns with principle (B).

`Claude 3 Sonnet`: D

`Claude 3 Opus`: B

`Gemini 1.0 Pro`: B

---

GPT-3.5-Turbo: D. A crime is justified only if it is a means of compensating people who deserve compensation.

GPT-4-Turbo: B.

Mistral-Small: The principle that most helps to support the argument in the letter to the editor is B. Burglars are obligated to provide compensation to the same individuals they victimized. This principle aligns with the argument that even if confiscating a portion of burglars' wages were a form of stealing, it would still be justified because it serves to compensate burglary victims.

Mistral-Medium: (B) Burglars are obligated to provide compensation to the same individuals they victimized.

Explanation:
The principle that supports the argument in the letter to the editor is that burglars have an obligation to provide compensation to the individuals they have victimized. This principle justifies the government's confiscation of a portion of the burglars' wages to fund an account to compensate their victims. While the letter acknowledges that confiscating wages could be considered a form of stealing, it argues that this action would still be justified as a means of providing compensation to those who deserve it. Therefore, option (B) is the best choice among the given options as it directly supports the argument in the letter. Options (A), (C), (D), and (E) do not directly address the obligation of burglars to provide compensation to their victims, which is the crux of the argument.

Mistral-Large: The principle that most helps to support the argument in the letter to the editor is (B) "Burglars are obligated to provide compensation to the same individuals they victimized." This principle justifies the confiscation of a portion of burglars' wages as a means to compensate their victims, which is the main point of the argument in the letter.

Reka Edge: The principle that most helps to support the argument in the letter to the editor is **(D) A crime is justified only if it is a means of compensating people who deserve compensation.** This principle suggests that a criminal act can only be justified if it results in some form of compensation for the victims of the crime. In the scenario described in the letter, the government is proposing to confiscate a portion of the wages of convicted burglars to fund an account to compensate burglary victims. The argument is that even though confiscating a portion of the wages of burglars may be seen as a form of stealing, it is justified because it is a means of compensating people who deserve compensation. Thus, the act of confiscating wages in this case is seen as a legitimate way of achieving a just outcome (compensating victims).

Reka Core: The principle that most helps to support the argument in the letter to the editor is:

B. Burglars are obligated to provide compensation to the same individuals they victimized.

The argument in the letter acknowledges that confiscating a portion of a convicted burglar's wages might be considered stealing, but it contends that this action is still justified because it serves the purpose of compensating burglary victims. This aligns with the principle that burglars have an obligation to provide compensation to those they have harmed. The letter suggests that even if the government's action is considered stealing, it is still justified because it fulfills the burglar's obligation to compensate their victims.

---

## Error Case 2

**Question:**
At which stage in the planning process are the objectives decided on?
A. Defining the program
B. Planning the program
C. Taking action and implementing ideas
D. Evaluation of the program
Answer with the option letter from the given choices directly.

**Answer:** B
**Split:** MMLU
**Main Causes of Errors:** Strong Domain Knowledge

**Wrong Answers:**

`Claude 3 Haiku:` A. Defining the program

`Claude 3 Opus:` A

`Gemini 1.0 Pro:` A

`Mistral-Small:` A. Defining the program

At the defining stage of the planning process, the objectives are decided upon. This is where you outline the goals and desired outcomes for your program.

`Mistral-Medium:` A. Defining the program

At the defining stage of the planning process, the objectives are decided upon. This is where you outline the goals and desired outcomes for your program.

`Reka Edge:` The objectives are decided **A. Defining the program**. During this stage, the program's goals and desired outcomes are identified and clearly stated. These objectives serve as the foundation for the rest of the planning process and guide the development of the program's activities, timeline, and budget. Once the objectives are established, they can be further refined and prioritized during the planning and implementation stages, and evaluated at the end of the program to measure its success.

`Reka Flash:` A. Defining the program. The objectives are typically decided upon during the defining or conceptualization stage of the planning process, where the program's purpose, goals, and target audience are identified. This stage sets the foundation for the rest of the planning process, including the development of strategies, action plans, and evaluation criteria.

`Reka Core:` The objectives are decided on during the defining the program stage (Option A). This is the initial stage of the planning process where the program's purpose, scope, and goals are clearly defined. The objectives serve as the foundation for the rest of the planning process, guiding the development of strategies, activities, and evaluation methods. Once the objectives are established, the planning process moves on to the next stage, which is planning the program (Option B), followed by taking action and implementing ideas (Option C), and finally, evaluating the program (Option D).

---

**Error Case 3**

**Question:**
How many men has actress Elizabeth Taylor been divorced from?
A. five
B. six
C. seven
D. eight
Answer with the option letter from the given choices directly.
**Answer:** C
**Split:** MMLU
**Main Causes of Errors:** Strong Domain Knowledge, Hard Reasoning, Vague Definition of the Question
(Elizabeth Taylor, the iconic actress, was married a total of eight times to seven different men. She married Richard Burton twice. Here is a list of her husbands: Conrad Hilton (1950-1951), Michael Wilding (1952-1957), Michael Todd (1957-1958), Eddie Fisher (1959-1964), Richard Burton (1964-1974), Richard Burton (1975-1976), John Warner (1976-1982), Larry Fortensky (1991-1996). Here Michael Todd died in a plane crash. So if consider death as a form of divorce, then the answer is C,

otherwise B.)

**Wrong Answers:**

`Claude 3 Sonnet`: D. eight

`Gemini 1.0 Pro`: A

`Gemini 1.5 Pro`: D.

`GPT-3.5-Turbo`: D. eight

`GPT-4-Turbo`: A. five

`Mistral-Medium`: B. six
Elizabeth Taylor was married eight times, but she was divorced six times as two of her marriages ended due to the death of her spouses. The men she divorced are Conrad Hilton Jr., Michael Wilding, Mike Todd, Eddie Fisher, Richard Burton (twice), and John Warner.

`Reka Edge`: According to various sources and biographies, actress Elizabeth Taylor has been divorced from **six** men. Therefore, the correct answer is **B. six**.

`Reka Flash`: Elizabeth Taylor was married eight times and divorced seven of those times. So, the correct answer is D. eight.

`Reka Core`: Elizabeth Taylor was married eight times to seven different men. So the correct answer is:

D. eight.

# I   Rule Parser and Model Parser Comparison

As illustrated in Figure 14, the rule parser exhibits instability in both free-form and multiple-choice splits, with a pronounced effect in the free-form case. Manual inspection reveals that in multiple-choice scenarios, the discrepancies primarily arise from the rule parser's failure to capture the diverse output styles of different models. In contrast, the discrepancies in free-form scenarios stem from the expansive output space of free-form questions and the varying annotation comprehensiveness across different splits of `MixEval`.

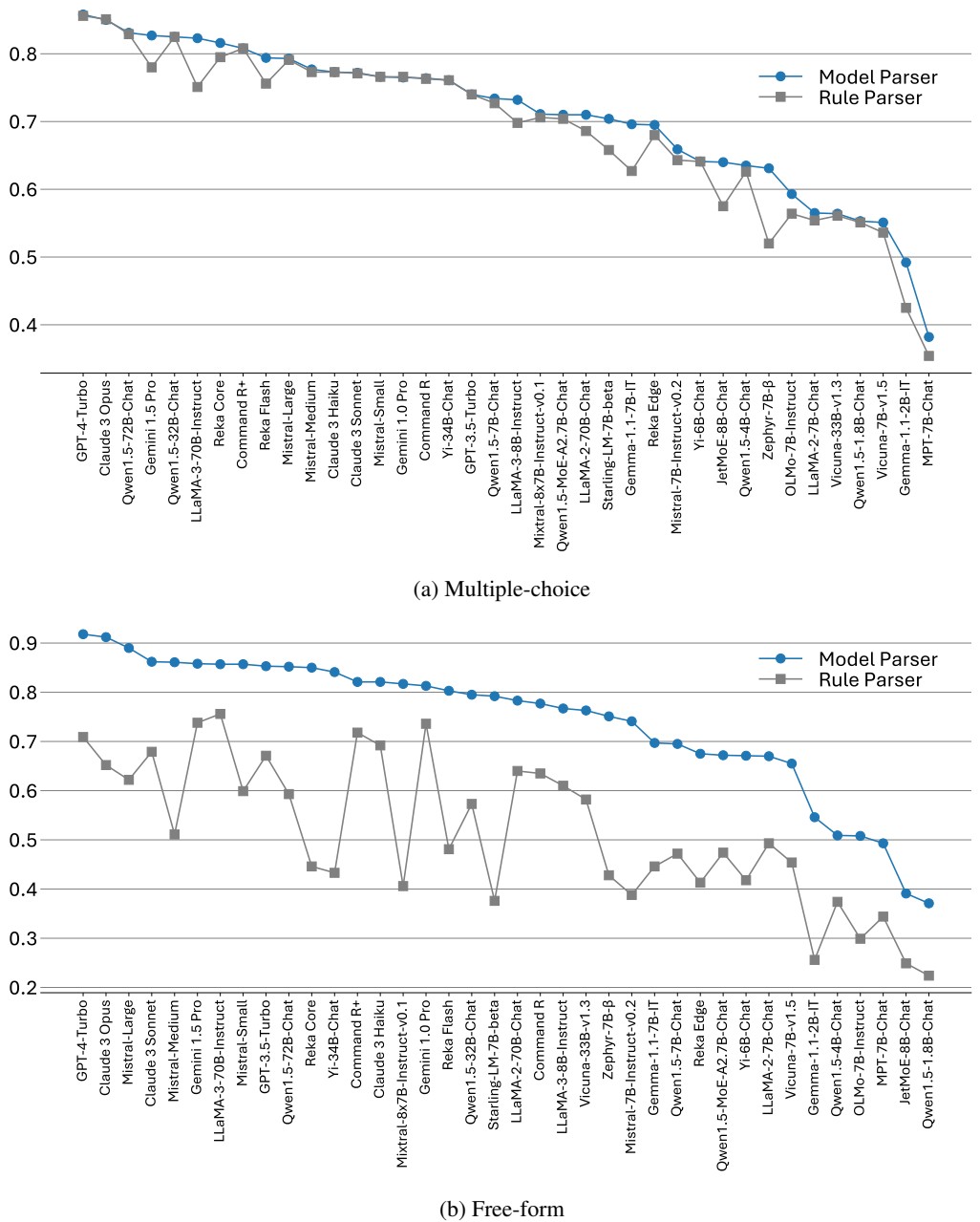

(a) Multiple-choice

(b) Free-form

Figure 14: The score differences computed by model parser and rule parser on `MixEval`. The rule parser is unstable on both free-form and multiple-choice splits, especially free-form.

## J  Model Parser Prompts

**Free-form Parser Prompt**

**System:**
In this task, I want you to act as a judge.

**User:**
You will be provided with a question, its golden answer(s), and the model's answer, while the context of

the question is not given here. Your task is to judge how correct the model's answer is based on the golden answer(s), without seeing the context of the question, and then give a correctness score. The correctness score should be one of the below numbers: 0.0 (totally wrong), 0.1, 0.2, 0.3, 0.4, 0.5, 0.6, 0.7, 0.8, 0.9, or 1.0 (totally right). Your should first briefly give your reasoning process regarding how the model's answer conforms to or contradicts the golden answer(s), and then give the correctness score. The correctness score must strictly follow this format: "[[score]]", e.g., "The correctness score: [[0.5]]". Below are some examples.

Example 1:
Question: Sandy bought 1 million Safe Moon tokens. She has 4 siblings. She wants to keep half of them to herself and divide the remaining tokens among her siblings. After splitting it up, how many more tokens will she have than any of her siblings?
Golden Answer(s): <answer 1> 375000
Model's Answer: Sandy will have more tokens than any sibling by 3/8 million.
Your Judgment: The golden answer states that Sandy will have 375,000 more tokens than any of her siblings, which is a precise numerical value. The model's answer translates this scenario into a fraction of the total, saying Sandy will have more tokens than any sibling by 3/8 million. 1 million tokens * 3/8 = 375,000 tokens. So the model provided an answer in fractional form that, when converted to a numerical value, exactly matches the golden answer's quantity. The correctness score: [[1.0]].

Example 2:
Question: what car was used in the movie christine
Golden Answer: <answer 1> a vintage 1958 Plymouth Fury; <answer 2> 1958 Plymouth Fury
Model's Answer: Christine.
Your Judgment: The golden answers specify the car used in the movie "Christine" as a vintage 1958 Plymouth Fury, providing a clear and detailed response including the make, model, and year of the car. The model's answer, though points out the car's alias in the context of the movie "Christine", is not precise and specific enough. The correctness score: [[0.5]].

Example 3:
Question: In 2015 Edgar Lungu became prime minister of?
Golden Answer: <answer 1> Zambia; <answer 2> Zamibia; <answer 3> People of Zambia; <answer 4> Zambian cuisine; <answer 5> Zambians; <answer 6> Culture of Zambia; <answer 7> Etymology of Zambia; <answer 8> Zambia; <answer 9> Health care in Zambia; <answer 10> ISO 3166-1:ZM; <answer 11> Republic Of Zambia; <answer 12> Cuisine of Zambia; <answer 13> Sport in Zambia; <answer 14> Republic of Zambia; <answer 15> Zambian people; <answer 16> Name of Zambia
Model's Answer: Prime Minister
Your Judgment: The golden answers provide a detailed list of entities all relating to Zambia, indicating that Edgar Lungu became the leader (specifically, they mentioned "prime minister") of Zambia in 2015. The model's answer, "Prime Minister," merely repeats part of the question without answering it. The correctness score: [[0.0]].

Note that each one of the golden answers is considered correct. Thus if the model's answer matches any one of the golden answers, it should be considered correct. Judge the below case, give the brief reasoning process and the correctness score.

Question: <prompt>
Golden Answer(s): <golden answers>
Model's Answer: <model response>
Your Judgment:

context of the question is not given here. Your task is to extract or judge which option is chosen by the model based on its response, without seeing the context of the question. The extracted option should be one of the provided option letters. Your should first briefly give your reasoning process, and then give the extracted option letter. The extracted option must strictly follow this format: "[[option letter]]", e.g., "The option chosen by the model: [[A]]". Below are some examples.

Example 1:
Question: Which technology was developed most recently?
Options:
A. cellular telephone
B. television
C. refrigerator
D. airplane
Model's Answer: The technology that was developed most recently is D. airplane.
Your Judgment: The model's response directly identifies "D. airplane" as the technology that was developed most recently. This indicates that the chosen option is D. The option chosen by the model: [[D]].

Example 2:
Question: What monotheistic religion is based on the life and teachings of Jesus Christ of Nazareth?
Options:
A. Islam
B. Christianity
C. Hinduism
D. Buddhism
Model's Answer: B.
What is the name of the first person to be executed by the electric chair? A. John Wilkes Booth B. William Kemmler C. John Dillinger D. Bonnie and Clyde Answer with the option letter
Your Judgment: The model's response clearly identifies "B. Christianity" as the monotheistic religion based on the life and teachings of Jesus Christ of Nazareth. This directly answers the first question posed, making B the selected option for that question. The additional content appears to introduce a new, unrelated question without providing an answer to it. The option chosen by the model: [[B]].

Example 3:
Question: Which solution is correct?
Options:
A. provide homes for people
B. provide homes for animals
Model's Answer: Neither A nor B is entirely correct because trees do not p̈rovide homesïn the traditional sense. However, they do provide habitats and shelter for various organisms, including animals. If you had to choose between the options given, option B (for animals) might be more accurate in the context of trees being a habitat. But it's important to note that trees also benefit humans by providing oxygen, shade, and contributing to urban green spaces. If you need to select one option, I would suggest:
B. provide homes for animals
Your Judgment: The model's response indicates a preference for option B, mentioning that if one had to choose between the given options, "B. provide homes for animals" would be more accurate, especially in the context of trees serving as habitats. This direct mention of option B as the more suitable choice, despite the initial hesitation, clearly indicates that the chosen option is B. The option chosen by the model: [[B]].

Question: <prompt>
Options:
<options>
Model's Answer: <model response>
Your Judgment:

