# OpenReview forum: "MixEval: Deriving Wisdom of the Crowd from LLM Benchmark Mixtures"
_NeurIPS.cc/2024/Conference — NeurIPS 2024 poster_

### Official Review · Reviewer_7ypd · 2024-07-02

**Soundness:** 3
**Presentation:** 3
**Contribution:** 3
**Rating:** 6
**Confidence:** 4

**Summary:**

The authors propose the MixEval to match the real-world human queries with existed benchmarks. MixEval is a two-stage benchmark reconstruction pipeline consisting of (1) wild query detection, and (2) grounding existing benchmarks in the mined queries. The authors match each crawled web user query with its most similar query in the benchmark pool and the corresponding ground truth answer to align benchmark queries with real-world queries. In order to improve the benchmark’s ability to distinguish strong models, the authors derive a challenging subset from MixEval, which is called MixEval-Hard.

**Strengths:**

1. The authors match each crawled web user query with its most similar query in the benchmark pool and the corresponding ground truth answer to align with the human perferences.
2. The experimental results demonstrate that the MixEval and MixEval-Hard are highly aligned with Chatbot Arena, outperform singular benchmarks.

**Weaknesses:**

1. In Section 3.2, the authors use dot product to match the query with the original benchmark, but do not explain how to use new queries in the test process and whether the queries are rewritten adaptively to match the ground truth answers. Why the answers from the original benchmark can be considered as the answers for the new queries?
2. In Figure 5, the description of the experimental setup is missing. It is not clear 0-shot or 5-shot is used in Figure 5, and how the inputs of "Mixed" and "Original" are formated for the models.
3. It is better to demonstrate the "User query" and "Ground truth-benchmark" processes are effective for constructing benchmarks through the ablation studies.

**Questions:**

1. Why can queries from different batches of the same distribution mitigate the contamination problem, instead of sampling queries from different distributions?
2. Why the performance of the Benchmark-level Mix in Figure 5 is lower than the average performance of the Mixed benchmarks?

**Limitations:**

The authors have discussed detailed limitations in the Appendix A.

---

> ### Author Rebuttal · Authors · 2024-08-06
>
> We thank the reviewer for finding this work solid and effective! Below are our responses to the concerns:
>
> ## Concern 1
> > In Section 3.2, the authors use dot product to match the query with the original benchmark, but do not explain how to use new queries in the test process and whether the queries are rewritten adaptively to match the ground truth answers. Why the answers from the original benchmark can be considered as the answers for the new queries?
>
> We thank the reviewer for the insightful comments! To clarify, after matching the web query with the original benchmark, **the original query will be dropped and we only use the questions and answers of the matched benchmark samples**. The benchmark mixture aims to map the real-world task distributions to the ground-truth-based benchmarks.
>
> We specified this in line 47-48, Figure 4 (graphic illustration), and Section 3.2 (notation illustration). We will carefully improve the related specifications in the later versions.
>
> ## Concern 2
> > In Figure 5, the description of the experimental setup is missing. It is not clear 0-shot or 5-shot is used in Figure 5, and how the inputs of "Mixed" and "Original" are formatted for the models.
>
> We thank the reviewer for pointing that out! **The setting for Figure 5 is the same as other experiments**: we use 0-shot and official, unified model input formatting; the correlation numbers of the "original" benchmarks are the same as Figure 1 and 10, whose settings are specified in Section E. We will make it clearer in the next version.
>
> ## Concern 3
> > It is better to demonstrate the "User query" and "Ground truth-benchmark" processes are effective for constructing benchmarks through the ablation studies.
>
> **We wish to highlight that we did the ablation study in Section 4.2–"MixEval outperforms both benchmark-level and uniform mixtures."** This ablation study aims to illustrate the effectiveness of the core step: benchmark mixture of MixEval. We compare the MixEval benchmark mixture with two other mixture schemes: Benchmark-level Mixture (ablating the sample-level mixture of MixEval) and Uniform Mixture (ablating both the sample-level and benchmark-level mixture of MixEval). Benchmark-level Mixture samples data points uniformly from each benchmark of the benchmark pool, with the benchmark size proportional to its split size in MixEval. In other words, its benchmark size distribution is the same as MixEval, while it's uniform inside each benchmark. The Uniform Mixture simply uniformly samples an equal number of data points from all benchmarks of the benchmark pool. Both methods yield significantly lower Arena Elo correlations than MixEval mixture, illustrating the effectiveness of our method.
>
> **Beyond that, we report the quantitative results of the whole query detection pipeline below (on our devised benchmarks) to illustrate the importance of the looped training.** Before being trained, a language model achieving high recall was chosen (Vicuna-33B), with 99.12% recall and 46.21% precision; The looped training (as described in Section 3.1) significantly improves the precision while maintaining the recall, illustrating the accuracy of the devised web query detection pipeline and the importance of the looped training.
>
> Table 2: The breakdown metrics of the web query detection pipeline.
> | Model           | Param | Pipeline Recall | Pipeline Precision | Pipeline F1 |
> |-----------------|-----------------|--------------|-----------|-----------|
> | Web Detector (initial) | 33B            | 99.12          | 46.21      | 63.03      |
> | Web Detector (trained)        | 33B            | 99.55          | 98.61      | 99.07      |
>
> Besides the above two ablations, the whole Section 4.2 has demonstrated the effectiveness of MixEval based on experimental results from multiple perspectives, which we believe has adequately illustrated the effectiveness.
>
> ## Concern 4
> > Why can queries from different batches of the same distribution mitigate the contamination problem, instead of sampling queries from different distributions?
>
> We thank the reviewer for this insightful comment!
>
> The main reason is that there is a significant difference between every two batches, as illustrated in the Table 2 of the submitted paper. Note that beyond **batch web query update**, we will also perform **source web query update**, which updates all the web user queries with the latest Common Crawl splits and it can be interpreted as updating queries from different distributions.
>
> **We analyzed the dynamism and contamination of MixEval in detail in the general response to all reviewers and AC (on the top of this page). Hope that will help solve this concern!**
>
> ## Concern 5
> > Why the performance of the Benchmark-level Mix in Figure 5 is lower than the average performance of the Mixed benchmarks?
>
> Because **benchmark-level mixture (Benchmark-level Mix)** is supposed to be worse than the **sample level mixture (Mixed)**.
> As we mentioned earlier, the "Benchmark-level Mix" is part of the ablation study, which samples data points uniformly from each benchmark of the benchmark pool, with the benchmark size proportional to its split size in MixEval. In other words, its benchmark size distribution is the same as MixEval, while it's uniform inside each benchmark. So such mixture scheme is supposed to perform worse than the "Mixed" benchmarks in Figure 5, which, as specified in line 238-239, performs the same sample-level mixture as MixEval. This performance gap arises from the fact that MixEval reconstructs real-world use cases more effectively than "Benchmark-level Mix".

---

> > ### Comment · Reviewer_7ypd · 2024-08-13
> > **Thank you for your response!**
> >
> > Thanks for the response. Looking forward to your final version.

---

### Official Review · Reviewer_u4BL · 2024-07-09

**Soundness:** 3
**Presentation:** 2
**Contribution:** 3
**Rating:** 5
**Confidence:** 4

**Summary:**

The paper reconstructs a new benchmark named MixEval by matching queries collected from the internet with existing benchmarks. This new benchmark aligns with the distribution of human preferences, reflecting the real distribution of queries on the internet. Additionally, considering the overlap and difficulty among different benchmarks, the paper introduces MixEval-Hard by recalculating model scores across various benchmarks. Both MixEval and MixEval-Hard can be updated quickly. Finally, the paper demonstrates that MixEval aligns better with Arena Elo compared to other benchmarks, and the mixing operation enhances the alignment of other benchmarks with Arena Elo.

**Strengths:**

1. The experiments are comprehensive, providing substantial results that validate the proposed method's ability to align well with the Arena Elo benchmark.
2. By utilizing queries from the internet to shift the distribution of existing benchmarks to reflect real human preferences, the approach helps prevent large models from overfitting on existing benchmarks. Additionally, the paper offers insights into the consistency between benchmarks, indicating that alignment results are influenced not only by query distribution but also by question difficulty and density.
3. An interesting visualization method is proposed, effectively reflecting the main aspects evaluated by the existing benchmarks.

**Weaknesses:**

1. Given that alignment with Arena Elo is used to measure the degree of alignment with human preferences, why not use data directly from Arena Elo when constructing MixEval, or even create a subset sampled from Arena Elo (considering Arena Elo directly reflects human preferences)?
2. To avoid models overfitting on existing benchmarks, the paper updates the benchmarks by drawing new queries from the internet. However, I believe overfitting occurs due to models overfitting on a fixed benchmark. Without changing the benchmark, the overfitting issue will not be resolved. Therefore, a method to determine if a benchmark is overfitted should be designed to add LLM benchmarks to the pool. Additionally, Table 2 does not demonstrate that this method alleviates overfitting.
3. Queries drawn from the internet might have already been seen by LLMs, so the retrieved samples from existing benchmarks might be simpler. The authors could exclude such factors during query extraction.
4. Some symbols are not explained, such as "j" in line 187, "tau" in the formula between lines 187 and 188, and "lambda" in line 189.

**Questions:**

Please see above.

**Limitations:**

Yes.

---

> ### Author Rebuttal · Authors · 2024-08-06
>
> We thank the reviewer for appreciating the comprehensive experiments, analysis, and insights of this work. Meanwhile, we understand the reviewer's concerns, which are also very important to us. Below we clarify:
>
> ## Concern 1
> > Given that alignment with Arena Elo is used to measure the degree of alignment with human preferences, why not use data directly from Arena Elo when constructing MixEval, or even create a subset sampled from Arena Elo (considering Arena Elo directly reflects human preferences)?
>
> We thank the reviewer for raising this concern. We think there may exist some misunderstandings here.
>
> As a warm reminder, we illustrated the reasons of using web queries in Section A.2 (FAQs). Below we reorganize Section A.2 as a response to this concern:
>
> First of all, it's worth noting that we are just taking Chatbot Arena as a measure of the capable model ranking of MixEval, **while our approach and data have nothing to do with Chatbot Arena**–our real-world user queries are solely detected from the web and the benchmark pool consists of the existing ground-truth-based benchmarks.
>
> More importantly, **our goal isn't to fit Chatbot Arena's distribution; we are trying to fit real-world task distributions instead**. As shown in Section 2.2, line 127, **Chatbot Arena queries are slightly biased towards technical users**. However, web queries are grounded in the largest human population (5.4 billion internet users) among the accessible query sources and thus being more representative for the real-world users (illustrated Section 2.2).
>
> Moreover, **the web queries are multi-modal**, meaning that the queries on the web do not only comprise the text-to-text user queries, they also feature other modalities, such as image-to-text, text-to-audio, etc. Such queries are not available on Chatbot Arena or other chat platforms if they do not support models with the corresponding inputs/outputs. Our detected queries of other modalities provide a proxy of building an upgraded MixEval with any-to-any modalities, which facilitates a better evaluation suite for the whole AI community. However, without web queries, it's impractical to get well-distributed any-to-any real-world queries within the community. We will release the multi-modal version soon.
>
> ## Concern 2
> > To avoid models overfitting on existing benchmarks, the paper updates the benchmarks by drawing new queries from the internet. However, I believe overfitting occurs due to models overfitting on a fixed benchmark. Without changing the benchmark, the overfitting issue will not be resolved. Therefore, a method to determine if a benchmark is overfitted should be designed to add LLM benchmarks to the pool. Additionally, Table 2 does not demonstrate that this method alleviates overfitting.
>
> We thank the reviewer for this insightful comment!
>
> **We analyzed the dynamism and contamination of MixEval in detail in the general response to all reviewers and AC (on the top of this page). Hope that will solve this concern!**
>
> As mentioned, although the contamination ratio of MixEval is comparatively low, we will indeed do benchmark contamination detection before putting it to the benchmark pool in the future versions of MixEval to further reduce the contamination. In addition, we will perform benchmark pool update to incorporate newly released ground-truth-based benchmarks to the benchmark pool, e.g., replacing MMLU with MMLU-pro, to mitigate contamination.
>
> ## Concern 3
> > Queries drawn from the internet might have already been seen by LLMs, so the retrieved samples from existing benchmarks might be simpler. The authors could exclude such factors during query extraction.
>
> Thanks for pointing that out! We will carefully add a pre-processing step accordingly in the future MixEval releases. And at the same time, we illustrate below that the described situation has **negligible impact** to the model evaluation.
>
> This concern can be converted to "We should exclude the benchmark queries leaked to the web from the extracted web queries to avoid introducing bias in the later benchmark mixture stage" (This conversion requires some reasoning. Due to the character limitation, we do not expand here and would be happy to discuss later.) **We argue that it will have negligible impact to the model evaluation**. Below we do a rough estimation about the ratio of the leaked benchmark queries in our extracted web queries.
>
> The benchmark pool has a size of 10^5-10^6. Suppose the web sentences has a size of 10^12-10^13 entries [1] and it has a query ratio of 6%, i.e., in 100 web sentences there are 6 valid web queries, then the web query quantity would be around 10^11-10^12. Thus, the ratio of benchmark entries in the web queries will only be around 10^-6. **Containing such a low proportion of leaked benchmark queries in our web queries will introduce negligible bias to the later benchmark mixture stage, because the amount of web queries we finally sample for benchmark mixture is only around 10^3-10^4, meaning it’s highly possible that there won’t be any leaked benchmark query in the final sampled web queries**.
>
> Besides, in the previous illustration of contamination and dynamism (Table 1 of this rebuttal, in the general response), it is shown that MixEval is relatively less contaminated (~10%) compared with other popular benchmarks, which can also be interpreted as "MixEval samples are less likely to have already been seen by LLMs".
>
> [1] Xue, Fuzhao, et al. "To repeat or not to repeat: Insights from scaling llm under token-crisis." Advances in Neural Information Processing Systems 36 (2024).
>
> ## Concern 4
> > Some symbols are not explained, such as "j" in line 187, "tau" in the formula between lines 187 and 188, and "lambda" in line 189.
>
> Thanks for pointing that out! These symbols are indeed under-specified, and we will carefully fix it in the next version. Here the j denotes the model's index; τ denotes the rejection threshold; λ denotes the scaling hyperparameter.

---

> > ### Comment · Reviewer_u4BL · 2024-08-12
> > **Thanks for the response!**
> >
> > Thank you for your response. The user's query is multimodal, so it is reasonable to use a mixed benchmark. My concern regarding the leakage issue has been partially resolved. Additionally, concerning Concern 3, although the queries in the benchmark are relatively rare on the entire web, it's possible that all queries on the web have been used as training data for the query-matching model. Therefore, I wonder if using these queries to match those in the benchmark might be more likely to match the polluted queries rather than with queries that the model has never seen. However, based on Table 1 in the final rebuttal response, it seems that the query matching does not introduce this bias. Thank you again for your response, and I will raise my score.

---

> > > ### Author Response · Authors · 2024-08-12
> > > **Thank you for raising score!**
> > >
> > > Thank you for your insightful comments and appreciation. Your support is very important to MixEval!

---

### Official Review · Reviewer_4BB9 · 2024-07-13

**Soundness:** 3
**Presentation:** 3
**Contribution:** 3
**Rating:** 6
**Confidence:** 3

**Summary:**

The paper introduces MixEval, a new benchmarking framework designed to overcome the limitations of traditional benchmarks and LLM-as-judge methods for evaluating LLMs. By leveraging web-mined user queries and matching them with existing benchmark queries, MixEval aims to offer a fast, efficient, and dynamic evaluation method that aligns closely with real-world human preferences. This framework promises significant cost and time savings and shows a high correlation with human preference leaderboards like Chatbot Arena.

**Strengths:**

1. This paper tackles the important evaluation problem timely. MixEval introduces a fresh approach to bridging the gap between real-world user queries and efficient evaluation. By using web-mined queries, it aims to better reflect actual user interactions.
2. The framework is efficient, could do dynamic updates and has high Correlation with Human Preferences.
3. The paper goes above and beyond by providing extensive analysis and comparison with other popular LLM benchmarks, giving valuable insights into the strengths and weaknesses of different evaluation methods.
4.  The paper is nicely written, with figures and tables well-organized. In particular, Figure 2 stands out for its clarity and effectiveness in presenting complex information.

**Weaknesses:**

1. Pipeline Brittleness: a. Web User Query Detection. The web user query detection phase has an overabundance of negative examples, which does little to help distinguish positive examples. b. Benchmark Mixture. Additionally, the sentence transformer used in the benchmark mixture phase has limited performance. Have you tried other sentence embedding models? Also it's unclear how many ground-truth LLM benchmarks can be accurately matched to mined queries.  c. The error rates for each module and the accumulated error rate are not provided.
2. Single-Language Focus: The all-mpnet-base-v2 model used in the framework is designed for English only, raising concerns about its adaptability to different linguistic and cultural contexts.

**Questions:**

1. Could you provide detailed qualitative and quantitive analysis of module Web User Query Detection and Benchmark Mixture?
2. It seems that the query of MixEval-Hard is longer than MixEval, is there any distribution difference between them?

**Limitations:**

See weaknesses.

---

> ### Author Rebuttal · Authors · 2024-08-06
>
> We thank the reviewer for recognizing MixEval as a timely work to the community! We are also grateful for your acknowledgment of the novelty, efficiency, thoroughness, and clarity of our work. Below are our responses to the concerns:
>
> ## Concern 1 & 3
> > Pipeline Brittleness: a. Web User Query Detection. The web user query detection phase has an overabundance of negative examples, which does little to help distinguish positive examples. b. Benchmark Mixture. Additionally, the sentence transformer used in the benchmark mixture phase has limited performance. Have you tried other sentence embedding models? Also it's unclear how many ground-truth LLM benchmarks can be accurately matched to mined queries. c. The error rates for each module and the accumulated error rate are not provided. Could you provide detailed qualitative and quantitive analysis of module Web User Query Detection and Benchmark Mixture?
>
> **Overabundance of negative examples** We controlled a balanced positive and negative samples for training by down-sampling the negative samples. The original specification may be a bit ambiguous, we will carefully fix this in the revised version later.
>
> **Weak retriever** It's worth noting that in our retrieval task, both the query and key are short sentences, as the queries are real-world user prompts and the keys are benchmark questions, therefore it does not require state-of-the-art embedding models. We tested with both all-mpnet-base-v2 and openai's text-embedding-ada-002, they showed negligible difference in our task (only 6 cases are significantly different over 500 retrievals). Besides, as shown in Figure 2, MixEval data points exhibit a perfect mapping from the original web queries (the lowest C-Dist), further illustrating its effectiveness.
>
> **Quantitative analysis of web query detection** We report the quantitative results of the whole detection pipeline below (on our devised benchmarks). Before being trained, a language model achieving high recall was chosen (Vicuna-33B), with 99.12% recall and 46.21% precision; The looped training (as described in Section 3.1) significantly improves the precision while maintaining the recall, illustrating the low error accumulation rate of the devised web query detection pipeline.
>
> Table 2: The breakdown metrics of the web query detection pipeline.
> | Model           | Param | Pipeline Recall | Pipeline Precision | Pipeline F1 |
> |-----------------|-----------------|--------------|-----------|-----------|
> | Web Detector (initial) | 33B            | 99.12          | 46.21      | 63.03      |
> | Web Detector (trained)        | 33B            | 99.55          | 98.61      | 99.07      |
>
> ## Concern 2
> > Single-Language Focus: The all-mpnet-base-v2 model used in the framework is designed for English only, raising concerns about its adaptability to different linguistic and cultural contexts.
>
> We thank the reviewer for pointing this out! This is indeed a good point. Though MixEval and MixEval-Hard are English-dominant (>95% English), keeping the whole pipeline as multi-lingual is beneficial. **Note that MixEval is dynamic and can be updated with time. We will replace the current retriever with a capable multi-lingual one in the future MixEval releases to further improve the distribution.**
>
> ## Concern 4
> > It seems that the query of MixEval-Hard is longer than MixEval, is there any distribution difference between them?
>
> Yes, they have some distribution differences, as they correspond to different difficulty levels. However, we have successfully controlled the distribution shift of MixEval-Hard with the rejection sampling mechanism introduced in Section 3.3. As shown in Figure 2, MixEval-Hard achieves a low C-Dist with the original web queries. The average length difference might arise from the fact that harder tasks tend to be longer in its average length, which aligns well with our commonsense.

---

### Official Review · Reviewer_yGD1 · 2024-07-16

**Soundness:** 4
**Presentation:** 4
**Contribution:** 3
**Rating:** 7
**Confidence:** 4

**Summary:**

This paper introduces MixEval, a new approach/benchmark to evaluate LLMs effectively in real-world scenarios. Traditional benchmarks often miss the comprehensiveness and subtlety of actual user queries, while existing methods like LLM-as-judge benchmarks are difficult to scale up. MixEval addresses these issues by using user queries mined from the web and aligning them with similar queries from established benchmarks.

The key advantage of MixEval is its efficiency and dynamic nature. It achieves a high correlation with Chatbot Arena (0.96 Spearman correlation) but requires only 6% of the time and cost compared to mainstream benchmarks like MMLU. MixEval can also be updated quickly, significantly reducing the risk of benchmark contamination. The paper also introduces MixEval-Hard for better differentiation among strong models. Through comprehensive analysis, the authors demonstrate that MixEval offers accurate, less biased, and more dynamic evaluations, providing a potential scalable solution for real-world LLM assessment.

**Strengths:**

1. Aligning web queries with mainstream benchmarks to simulate real-world user preferences is a novel approach. The authors cleverly transform the challenging task of evaluating open-ended queries into a benchmark mixture with groundtruth-based results, which is an interesting idea.
2. MixEval is an effective alternative to ChatBot Arena as it can scale up, dynamically update, and has lower costs.
3. The authors provide a comprehensive analysis of various LLMs on MixEval, demonstrating high correlations with ChatBot Arena.

**Weaknesses:**

**Major Issues**:
1. MixEval dynamically updates by mixing popular benchmarks (e.g., MMLU, BoolQ, GSM8K), which may not mitigate contamination. Most of these benchmarks are saturated and suffer from contamination. Although the authors claim they will "dynamically expand our benchmark pool with newly released benchmarks to further enhance the mixed benchmark distribution," this does not address the root issue of contamination.

**Minor Issues**:
1. While the topic distributions in Figure 2 are impressive, most of these benchmarks were not designed to simulate human preferences, hence the skewed topic distributions may be intentional for evaluating specific tasks, domains, or capabilities. Therefore, the section title "LLM Benchmarks are Biased from Realistic User Queries and Preferences" seems biased, implying these benchmarks are meant to measure human preferences.
2. Another potential issue is the assumption that ChatBot Arena serves as the groundtruth for calculating correlations. As noted by the authors in lines 127-133, ChatBot Arena itself is biased. Therefore, even though MixEval has a larger user base, it might overfit to ChatBot Arena to some extent, making it more of an alternative rather than a more accurate evaluation of user preferences.

**Questions:**

I am a bit confused about Table 2. The authors briefly mention "periodically update" in the main text, but I did not fully understand the details of this update. Could you explain this further?

**Limitations:**

In the appendix, the authors address some limitations in a Q&A format, such as the potential biases in MixEval. My views on limitations are reflected in the weaknesses above.

---

> ### Author Rebuttal · Authors · 2024-08-06
>
> We thank the reviewer for finding this work novel, effective, comprehensive, and solid! Below are our responses to the concerns:
>
> # Major Issue:
>
> ## Concern 1
> > MixEval dynamically updates by mixing popular benchmarks (e.g., MMLU, BoolQ, GSM8K), which may not mitigate contamination. Most of these benchmarks are saturated and suffer from contamination. Although the authors claim they will "dynamically expand our benchmark pool with newly released benchmarks to further enhance the mixed benchmark distribution," this does not address the root issue of contamination.
>
> We thank the reviewer for this insightful comment!
>
> **We analyzed the dynamism and contamination of MixEval in detail in the general response to all reviewers and AC (on the top of this page). Hope that will solve this concern!**
>
>
> # Minor Issues:
>
> ## Concern 2
> > While the topic distributions in Figure 2 are impressive, most of these benchmarks were not designed to simulate human preferences, hence the skewed topic distributions may be intentional for evaluating specific tasks, domains, or capabilities. Therefore, the section title "LLM Benchmarks are Biased from Realistic User Queries and Preferences" seems biased, implying these benchmarks are meant to measure human preferences.
>
> Thanks for pointing that out, this is an interesting topic to discuss!
>
> First of all, as indicated in the section title, what we care about are **real-world use cases** and **user preferences**, instead of solely the user preferences. "Real-world use cases" depicts the distribution of tasks, while "user preferences" depicts the grading process.
>
> **We believe the core principle of doing model evaluation is to evaluate them as they will be used in the real world.** As a result, the evaluations should be designed based on this principle. (We also illustrated this in Section A.1 "Why are real-world human queries and preferences important?")
>
> **However, most of the existing evaluations are developed in a way that is based on the interest of its creators instead of the real-world users**, e.g., GSM8K was created to measure models' mathematical reasoning abilities. Is mathematical reasoning something that we should measure? Yes, but we are not sure how important it is. Hence, Section 2 aims to measure the significance of existing evaluations by (1) compare their distributions with real-world use case distribution and (2) show their correlations with large-scale human preferences. "LLM Benchmarks are Biased from Realistic User Queries and Preferences" is the conclusion we got after rigorous analysis.
>
> It is important to note that Section 2 did not aim to completely criticize the existing evaluations, because they are still a good measure for specific tasks, domains, or capabilities. Section 2 is only serving as a meta-evaluation to the community: (1) it tells people that the results obtained from these evaluations may not generalize to real-world use cases, as they show a deviated task distribution and limited correlation with real-world use cases, (2) it visualizes the task distributions of different evaluations to help people select the correct benchmarks for the abilities they want to measure–e.g., it shows that you shouldn't take WinoGrande or DROP as a general-purpose benchmark, as their tasks only focus on a very small range of topics.
>
> ## Concern 3
> > Another potential issue is the assumption that ChatBot Arena serves as the groundtruth for calculating correlations. As noted by the authors in lines 127-133, ChatBot Arena itself is biased. Therefore, even though MixEval has a larger user base, it might overfit to ChatBot Arena to some extent, making it more of an alternative rather than a more accurate evaluation of user preferences.
>
> We thank the reviewer for this insightful comment! As illustrated by the footnote of the first page, the Chatbot Arena leaderboard is indeed not the sole indicator of human preference, but it currently serves as the only gold standard large-scale human preference benchmark within the community. Therefore, we could not find a better proxy of real-world user preferences than Chatbot Arena to compared with. Because of this, we didn't claim that MixEval provides a better approximation of real-world human preference than Chatbot Arena (it might be better or not); instead, MixEval is providing an efficient and low-biased evaluation that can reflect real-world use cases.
>
> ## Concern 4
> > I am a bit confused about Table 2. The authors briefly mention "periodically update" in the main text, but I did not fully understand the details of this update. Could you explain this further?
>
> Sure! As illustrated in lines 51-53 and Section 3.4 of the submitted paper, we update the data points of MixEval via (1) **batch web query update** (sampling different web queries batches from the crawled web queries), (2) **source web query update** (updating all the web queries with the latest Common Crawl) or (3) **benchmark pool update** (incorporating new ground-truth-based benchmarks to the benchmark pool). Since the mechanism of MixEval is to match web queries with benchmark pool samples, the above three updating methods refreshes both the web queries (the first and the second method) and benchmark pool samples (the third method). We will specify it more clearly in the next version.

---

> > ### Comment · Reviewer_yGD1 · 2024-08-11
> > **Thank You for the Response!**
> >
> > Thank you for your response and the additional experiments, which have led me to increase the score from 6 to 7. I agree that MixEval can mitigate contamination, but I still do not believe it fundamentally solves the problem, as it still heavily relies on existing benchmarks. I disagree with the authors' explanation for Concern 2, as many benchmarks were not originally designed to measure human preferences, and human preferences are just one aspect of the many evaluations for LLMs (although a very important one). Technical reports for all LLMs release results from numerous benchmarks to give the community a comprehensive understanding of their capabilities. Calculating correlations between all these benchmarks and human preferences introduces some potential biases, such as assuming they should align with human preferences, which is not necessarily the case.
> >
> > Despite these issues, which I consider minor, I personally recommend the acceptance of this paper. Good luck!

---

> > > ### Author Response · Authors · 2024-08-12
> > > **Thank you for your appreciation!**
> > >
> > > Thank you once again for recognizing MixEval and for your insightful comments. We will incorporate the feedback from this rebuttal to enhance future releases and revisions.

---

### Author Rebuttal · Authors · 2024-08-04

We thank all the reviewers for their valuable feedback. We identify that the main concern among reviewers is about the contamination of MixEval. **Therefore, we conduct additional contamination analysis and provide the general response here.**

# High-Level Takeaways

1. **Low Natural Contamination**: MixEval and MixEval-Hard demonstrate a low natural contamination ratio, as detailed in the Table 1 of this rebuttal (below).
2. **Purpose of Model Evaluations**: Model evaluations typically serve two main purposes: **developer self-validation** and **leaderboard competition**. For self-validation, contamination of MixEval is not a concern. For leaderboard competition, MixEval effectively mitigates contamination through **batch web query updates**, **source web query updates**, and **benchmark pool updates**, demonstrating its high dynamism compared with traditional ground-truth benchmarks.
3. **Broader Contributions**: While dynamism is a key feature of MixEval, the contributions of this work extend significantly beyond this aspect, as elaborated in lines 60-73 of the submitted paper.

# Detailed Justification

When does contamination affect model evaluations? Generally, model evaluations serve two purposes: **developer self-validation** and **leaderboard competition** [1].

## Self-Validation
For self-validation, MixEval is effective regardless of contamination levels, as **developers aim to exclude evaluation data from training data to ensure accurate testing**. At this stage, they will not deliberately overfit the evals. In addition, they usually employ contamination detection to remove leaked evaluation data from the training data, enhancing the generalizability of the internal evaluation pipeline. MixEval, being efficient and low-biased, accurately mirrors real-world use cases, making it ideal for rapid model iteration.

## Leaderboard Competition
Conversely, in leaderboard competitions, some contamination is inevitable due to the static nature of ground-truth-based benchmarks. **Developers may not rigorously exclude evaluation data from training, or may even include it intentionally to improve leaderboard rankings**. Below, we demonstrate that MixEval can mitigate both natural and deliberate contamination..

**We wish to highlight that MixEval is only mitigating the contamination instead of solving it completely, as mentioned on line 50, 207, and Figure 4 of the submitted paper.**

### Natural Contamination
**For natural contamination, MixEval mitigates it via benchmark mixture.** According to Table 1 of [2], contamination levels of existing web benchmarks range from 1.1% to 40.6%. Generally, more popular benchmarks exhibit higher contamination. For example, MMLU shows a relatively high contamination ratio (24.3%), yet remains crucial to the community and our benchmark pool. MixEval addresses this by mixing popular benchmarks with less contaminated ones smartly (e.g., CommonsenseQA), thus reducing the natural contamination ratio. Note that MixEval can be updated with time, and we will include contamination detection in future releases to further minimize contamination.

We use the contamination detector in [2] to detect natural contaminations of benchmarks by searching the ratio of benchmark data existing on the web. As shown in the table below, MixEval and MixEval-Hard exhibit a lower contamination ratio compared with popular benchmarks such as MMLU due to the smoothing effect mentioned above. Overall, MixEval achieves the highest correlation with real-world use cases while staying at a low natural contamination ratio and high efficiency.

Table 1: Contamination ratio of different benchmarks.
| Dataset        | Split | #Total | #Input-only Contamination | #Input-and-label Contamination |
|----------------|-------|--------|---------------------------|-------------------------------|
| ARC_c          | Test  | 1172   | 53 (4.5%)                 | 283 (24.1%)                   |
| CommonsenseQA  | Dev   | 1221   | 3 (0.2%)                  | 17 (1.4%)                     |
| Winogrande     | Dev   | 1267   | 0 (0.0%)                  | 14 (1.1%)                     |
| C-Eval         | Dev   | 1346   | 69 (5.1%)                 | 547 (40.6%)                   |
| Hellaswag      | Dev   | 10042  | 46 (0.5%)                 | 1201 (12.0%)                  |
| MMLU           | Test  | 13987  | 678 (4.8%)                | 3399 (24.3%)                  |
| **MixEval**        | Test  | 4000   | 129 (3.2%)                | 414 (10.4%)                   |
| **MixEval-Hard**   | Test  | 1000   | 39 (3.9%)                 | 106 (10.6%)                   |

### Deliberate Contamination
**For deliberate contamination, MixEval mitigates it by dynamically updating web user queries and the benchmark pool.** If model developers deliberately overfit evals, contamination is nearly impossible to fully eliminate. Even with dynamic systems like Chatbot Arena, evaluations can still be hacked, e.g., fitting on LMSys user data or hiring biased workers. Developers may hack MixEval by **(1) directly fitting on MixEval data**, or **(2) fitting the benchmark pool**. We address method (1) by periodically updating MixEval data points through "**batch web query update**" (sampling new web query batches from the crawled web query pool) or "**source web query update**" (updating the whole web query pool with the latest Common Crawl), and then perform benchmark mixture. Table 2 of the submitted paper shows it effectiveness, demonstrating significant differences between MixEval versions. Method (2) is tackled by "**benchmark pool update**", incorporating new ground-truth benchmarks in the community, e.g., replacing MMLU with MMLU-pro.

We will add this discussion to Section A as another FAQ to strengthen people's understanding regarding the dynamism of MixEval.

Reference:

[1] Fourrier, Clémentine. "Let's Talk About LLM Evaluation."

[2] Li, Yucheng. "An open source data contamination report for llama series models.".

---

> ### Author Response · Authors · 2024-08-10
>
> Dear Reviewers,
>
> Thank you for your dedication and time spent reviewing our paper.
>
> As the discussion period draws to a close, we wish to remind you of the approaching deadline.
>
> Please feel free to raise any points or questions about the paper that may need additional clarification.
>
> We deeply appreciate your valuable feedback.
>
> Thank you once again.

---

### Decision · Program_Chairs · 2024-09-25

**Decision:**

Accept (poster)

**Comment:**

This paper proposes a new benchmark designed to better align LLM evaluations with real-world user queries called MixEval. MixEval matches web-mined queries to existing benchmarks, aiming to offer a fast, efficient, and dynamic evaluation method.

The reviewers agreed that the proposed approach was novel and efficient and that the paper was well-written and provided a comprehensive analysis of the experimental results.

The reviewers also identified a number of areas where the paper could be improved.  Some of their suggestions were related to the robustness of the approach, questions around generalizability beyond English, and concerns about things like contamination, bias, and overfitting. The authors are strongly encouraged to carefully consider this feedback and address it to the extent possible in future versions of the paper.